# INVERSE CONSTITUTIONAL AI:
# COMPRESSING PREFERENCES INTO PRINCIPLES

**Arduin Findeis**[*]
University of Cambridge
Cambridge, UK

**Timo Kaufmann**[†]
LMU Munich, MCML Munich
Munich, Germany

**Eyke Hüllermeier**
LMU Munich, MCML Munich
DFKI, Kaiserslautern, Germany

**Samuel Albanie**
London, UK

**Robert Mullins**
University of Cambridge
Cambridge, UK

## ABSTRACT

Feedback data is widely used for fine-tuning and evaluating state-of-the-art AI models. Pairwise text preferences, where human or AI annotators select the "better" of two options, are particularly common. Such preferences are used to train (reward) models or to rank models with aggregate statistics. For many applications it is desirable to *understand* annotator preferences in addition to modelling them — not least because extensive prior work has shown various unintended biases in preference datasets. Yet, preference datasets remain challenging to interpret. Neither black-box reward models nor statistics can answer *why* one text is preferred over another. Manual interpretation of the numerous (long) response pairs is usually equally infeasible. In this paper, we introduce the *Inverse Constitutional AI* (ICAI) problem, formulating the interpretation of pairwise text preference data as a compression task. In constitutional AI, a set of principles (a *constitution*) is used to provide feedback and fine-tune AI models. ICAI inverts this process: given a feedback dataset, we aim to extract a constitution that best enables a *large language model* (LLM) to reconstruct the original annotations. We propose a corresponding ICAI algorithm and validate its generated constitutions quantitatively based on annotation reconstruction accuracy on several datasets: (a) synthetic feedback data with known principles; (b) AlpacaEval cross-annotated human feedback data; (c) crowdsourced Chatbot Arena data; and (d) PRISM data from diverse demographic groups. As an example application, we further demonstrate the detection of biases in human feedback data. As a *short and interpretable representation* of the original dataset, generated constitutions have many potential use cases: they may help identify undesirable annotator biases, better understand model performance, scale feedback to unseen data, or assist with adapting AI models to individual user or group preferences. We release the source code for our algorithm and experiments at `https://github.com/rdnfn/icai`.

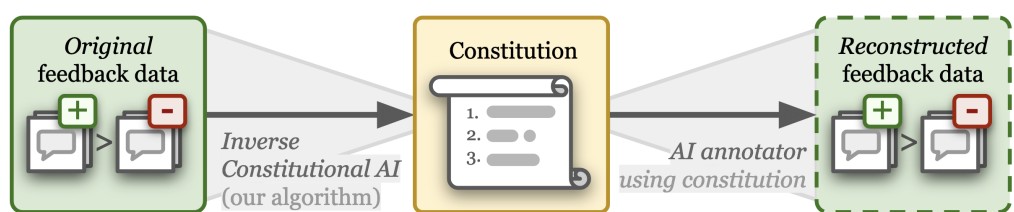

Figure 1: **The *Inverse Constitutional AI* problem.** Starting with pairwise preference feedback data, we derive a set of natural language principles (a *constitution*) that explain the preferences. For validation, we reconstruct the original preferences with an LLM judging according to the generated constitution. The constitution represents a (highly compact) compression of the preferences.

[*]`arduin.findeis@cst.cam.ac.uk`; [†]`timo.kaufmann@ifi.lmu.de`

# 1 INTRODUCTION

State-of-the-art AI models, in particular *large language models* (LLMs), rely heavily on human feedback for training and evaluation. This feedback, often in the form of *pairwise text preferences*, is crucial to assess advanced capabilities, which are hard to evaluate automatically. Pairwise text preferences typically consist of a *prompt*, two *model responses* and an *annotation* selecting the "better" response. Strategies for training on such data have seen widespread adoption, with notable examples including *reinforcement learning from human feedback* (RLHF) (Ouyang et al., 2022; Stiennon et al., 2020) and *direct preference optimization* (DPO) (Rafailov et al., 2023). Beyond training, pairwise text preferences are also widely-used for evaluating LLMs, such as in the *Chatbot Arena* (Chiang et al., 2024), where crowdsourced preferences determine rankings — possibly reflecting the complexity of human preferences better than conventional benchmarks (Xu et al., 2023).

However, interpreting such pairwise data is challenging: describing *what exactly* we train a model to do when applying RLHF with a large number of preference pairs is hard. Similarly, understanding *why* a model is ranked higher in a pairwise data-based leaderboard remains difficult. Yet, understanding such data is critical: human feedback is not without its flaws. Systematic biases in human judgement have been documented extensively in the psychology literature (Tversky & Kahneman, 1974). Perhaps unsurprisingly, biases have been similarly observed in the human feedback used to guide and evaluate LLMs. For example, human annotators have been observed to sometimes favour *assertiveness* (Hosking et al., 2024) or *grammatical correctness* (Wu & Aji, 2023) over truthfulness.

Feedback data with unintended biases can be problematic: when used for fine-tuning, biased data may lead to models that exhibit the same biases. Similarly, leaderboards based on biased data will *favour misaligned models* (Dubois et al., 2023; 2024). As such, understanding the implicit rules and biases guiding annotators of feedback data is valuable. To date, however, few general tools exist to detect biases in pre-existing preference data at scale. Prior work detecting biases usually either requires specially collected data or focuses on biases easy to programmatically measure (e.g. length bias (Dubois et al., 2024)) — difficult to extend to pre-existing datasets or less controlled settings.

In this paper, we propose a novel general approach to understanding preference corpora: *Inverse Constitutional AI* (ICAI). We make the following contributions:

1. **Formulating the *Inverse Constitutional AI* (ICAI) problem.** In Constitutional AI (Bai et al., 2022b), a set of principles (or *constitution*) is used to provide feedback and fine-tune language models. ICAI inverts this process: given a dataset of human or AI feedback, we seek to compress the annotations into a set of principles that enable *reconstruction* of the annotations (Figure 1).

2. **Proposing an initial ICAI algorithm.** We introduce a first ICAI algorithm that generates a set of principles based on a feedback dataset. We validate the constitutions generated by our algorithm based on their ability to help reconstruct feedback. Given the complexity of human judgement, the constitution necessarily represents a "lossy", non-unique compression of the feedback data. Nevertheless, the interpretable nature of the principles may enable a number of promising downstream use cases: (a) highlighting potential issues in preference data; (b) creating interpretable reward models; (c) scaling human-annotated evaluation to new models and use cases; and (d) generating personal constitutions for customized model behaviour.

3. **Providing experimental results and case studies.** We test our approach experimentally on four datasets: (a) we first provide a proof-of-concept on *synthetic data* with known underlying principles; (b) we then demonstrate applicability to human-annotated data on the *AlpacaEval dataset* (Dubois et al., 2023); (c) we showcase applicability to interpreting individual user preferences via *Chatbot Arena Conversations* data (Zheng et al., 2023); (d) we investigate the use-case of bias detection on different datasets; and finally (e) we demonstrate our method's ability to help interpret differing group preferences on *PRISM* data (Kirk et al., 2024). We demonstrate the highly sample-efficient generation of personalised constitutions with human-readable and editable principles. We release our code at `https://github.com/rdnfn/icai`.

## 2 THE INVERSE CONSTITUTIONAL AI PROBLEM

Given a set of pairwise preference feedback, the *Inverse Constitutional AI* (ICAI) problem is to generate a corresponding *constitution* of natural-language principles that enable an LLM annotator

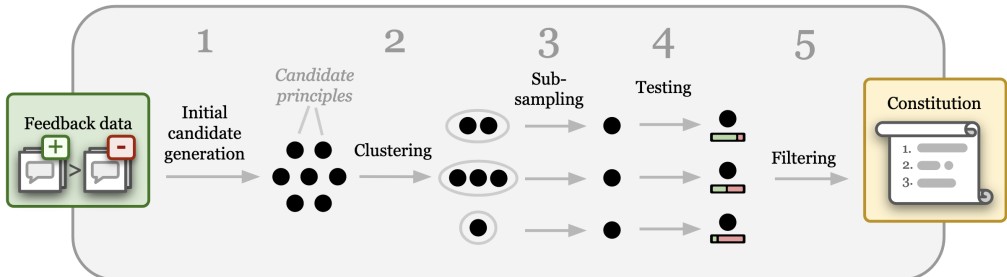

Figure 2: **Overview of our *Inverse Constitutional AI* (ICAI) algorithm.** Given a dataset of pairwise comparisons, in Step 1 candidate principles are *generated* using an LLM. In Step 2, these principles are *clustered* using an embedding model. In Step 3, similar principles are deduplicated by *sampling* one principle per cluster. In Step 4, each principle is tested to evaluate its ability to help an LLM reconstruct the original annotations. Finally, in Step 5, the principles are *filtered* according to the testing results, and a set of filtered principles are returned as the final *constitution*. Optionally, a final step of additional clustering and subsampling can follow to ensure diverse principles.

to reconstruct the original preferences as well as possible. Formally, we seek to find

$$\arg\max_{c} \{ \text{agreement}(p_o, p_M(c)) \text{ s.t. } |c| \leq n \}, \tag{1}$$

where $p_o$ are the original preferences and $p_M(c)$ are *constitutional* preferences over a pairwise preference corpus $T$, generated by LLM $M$ using the constitution $c$. The constitution is constrained to consist of up to $n$ human-readable natural language principles. Agreement is defined as the percentage of constitutional preferences $p_M(c)$ identical to the original preferences $p_o$. A constitution with high agreement can help interpret a preference dataset to gain insight into the underlying annotator preferences and biases. The constitution may also be used for future preference synthesis, with an interpretable and editable set of principles.

## 3 METHOD

We propose a first *Inverse Constitutional AI* (ICAI) algorithm, outlined in Figure 2, consisting of five main steps: *principle generation*, *principle clustering*, *principle subsampling*, *principle testing*, and *principle filtering*. In the following, we describe each step in detail.

**Step 1: Principle generation.** We extract candidate principles using an LLM with access to the feedback data. The principles are generated on a per-comparison basis: an LLM is prompted with a pair of texts and corresponding preference, and then asked to propose principles that explain the preference (prompts in Appendix G.1). The generated principles are in the form of natural language instructions that inform preference decisions (e.g., "select the more polite output"). We generate a large number of candidate principles using multiple (by default 2) generation prompts and multiple principles per prompt to cover a wide range of potential rules. The generation prompts affect the type of principles that get generated and tested (e.g., specific/general, positive/negative rules).

**Step 2: Principle clustering.** Since the first step generates a large number of candidate principles independently, almost identical principles may be generated multiple times. We use $k$-means-based clustering on embeddings to identify principles that are similar for merging. The parameter $k$ determines the number principles considered downstream and thus affects overall computational cost.

**Step 3: Principle subsampling.** In the third step, we deduplicate the principles by randomly sampling one principle per cluster, leading to a diverse set of remaining principles.

**Step 4: Principle testing.** The fourth step evaluates the generated principles' ability to help an LLM reconstruct the original annotations. We prompt the LLM with the generated principles to determine the 'votes' each principle casts across comparisons, which we then compare to the true annotations (see Appendix G.2). We parallelize this step, prompting the LLM with a pair of texts and multiple principles to provide a separate response (first preferred, second preferred, not relevant)

for each principle. This parallelization reduces the token requirements compared to separate testing. While LLMs can exhibit anchoring effects when predicting multiple labels in one output (Stureborg et al., 2024), we hypothesize this effect is less pronounced for relative preferences and our experimental results indicate sufficient reliability on our datasets. We compare these results to the original labels and count the correct, incorrect, and not relevant labels for each principle separately, thereby identifying principles that help the LLM to correctly annotate the dataset.

**Step 5: Principle filtering.** Finally, the principles are filtered based on the results of the previous testing step. We only keep principles that improve the reconstruction loss, while discarding principles that do not help or even hinder the reconstruction. We further discard principles that are marked as relevant on less than $x\%$ of the data (default $10\%$), to avoid overly specific principles that do not generalize. We order the principles according to their net contribution to correctly annotating the dataset (*#correct − #incorrect annotations*). We then select the top $n$ principles according to this order (see Appendix F.5 for further discussion). Optionally, to increase principle diversity, we cluster the top $m (> n)$ principles into $n$ clusters as before, and subsample the highest ordered principle from each cluster.[1] The final ordered list of principlesis returned as the *constitution*.

**Inference.** Given a constitution, we can validate its ability to "explain" the original feedback dataset. We do this validation using AI annotators prompted with the constitution, an approach pioneered by Bai et al. (2022b) and commonly referred to as *constitutional AI*. Notably this method leaves room for interpretation of the constitution by the AI annotator, as the constitution may be ambiguous, contradictory or incomplete, but results may also vary depending on the exact prompt and constitution. We build on the *AlpacaEval* framework (Li et al., 2024c) for our constitutional annotators. This inference using constitutional AI annotators enables quantitatively testing the validity of our generated constitutions and their ability to explain the data while also enabling downstream use cases, such as personalized preference models.

# 4 EXPERIMENTS

We conduct experiments on four datasets: (1) *synthetic data* to demonstrate the basic functionality of our algorithm, (2) human-annotated *AlpacaEval data* to demonstrate the applicability of our algorithm to real-world data, (3) *Chatbot Arena data* to illustrate the application of our algorithm to infer individual user preferences, and (4) *PRISM* data to showcase interpreting group preferences with our algorithm. Full dataset details are available in Appendix A. We primarily use two models from OpenAI: GPT-3.5-Turbo and GPT-4o. Example constitutions in all figures were chosen for illustrative purposes. We provide more constitutions, experiment details (including numerical results), and model details in Appendices D, F and H, respectively.

**Annotators.** We use annotators from the *AlpacaEval* framework (Li et al., 2024c) as baselines that have been shown to strongly correlate with human preferences (referred to as *Default* annotators, see Appendices F.8 and G.4). To evaluate constitution effectiveness, we create custom prompts that ask the model to annotate according to principles in a constitution (see Appendix G.3). The Default annotators cannot adjust to different datasets, performing poorly when preferences deviate from its default preferences. In contrast, our constitutional annotators are able to adapt, a key advantage of our approach. We show random baseline performance as a grey dashed line at $50\%$ in all plots. To contextualize ICAI's performance, we compare it against additional baselines: a flipped Default annotator, a (fine-tuned) reward model, and an annotator based on PopAlign (Wang et al., 2024) hypothesizing principles during inference. Details are in Appendix F.8, summarized here. Non-adaptive baselines (pretrained reward model, (flipped) Default) perform well on some datasets but fail to adjust to all. The fine-tuned reward model adapts partially but underperforms our constitutional annotator in our low-data scenario. A custom-trained reward model with extensive data may surpass our method but would require significant resources and lack interpretability.

## 4.1 PROOF-OF-CONCEPT: SYNTHETIC DATA

We first apply our algorithm to three *synthetic datasets* created according to known rules crafted to be aligned, unaligned, and orthogonal to the preferences internalized by the base LLM. Each dataset

---

[1]We found it important not to be too restrictive with the number of clusters in Step 2, as good principles may never be tested. More clusters can lead to duplicates in the tested rules, necessitating another filtering step.

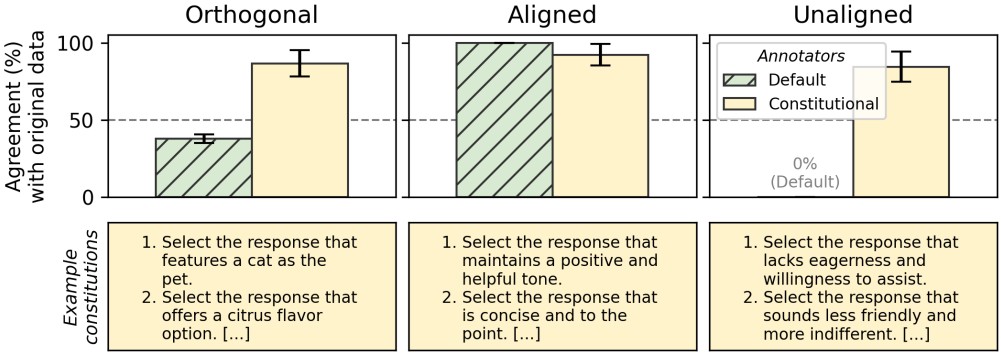

Figure 3: Results on synthetic data. **Our constitutional annotators can reconstruct a variety of preferences using limited data and without fine-tuning.** We demonstrate our algorithm's adaptability on three synthetic datasets: one *orthogonal* to the base LLM's learned preferences, one *aligned* with those preferences and one *unaligned* with them. We generate constitutions for each and report agreement with the original data of a *default* LLM annotator and a *constitutional* annotator (prompted with a constitution). Our constitutions notably improve agreement in the orthogonal and unaligned cases and retain high agreement in the aligned case, albeit with more variance. Our method's ability to detect biases is illustrated by the example constitution in the unaligned case. Plots show mean and standard deviation (6 seeds) using GPT-3.5-Turbo.

is generated from three principles, with 10 pairs per principle, resulting in 30 pairs per dataset. We provide an overview of these datasets here, with further details in Appendix I.

**Orthogonal.** This dataset is based on principles intended to be neither supported nor opposed by humans or language models on average. In particular, we create a dataset based on three principles: "prefer cats over dogs", "prefer green over blue color", and "select lemon over raspberry ice-cream".

**Aligned and unaligned.** The aligned dataset uses preferences generally accepted by humans and (especially) language models. Our dataset follows three principles: "select truthful over factually incorrect answers", "select helpful over useless answers", "select polite over impolite answers". The unaligned dataset flips these annotations, creating a dataset that a default LLM annotator mostly disagrees with.

**Results.** In Figure 3, we compare *default* annotators (prompted to select the "best" output) to *constitutional* annotators (prompted with a generated constitution). We find that constitutional annotators reconstruct original annotations better in the orthogonal and unaligned datasets, and keep high agreement in the aligned case (which offers limited room for improvement). These results indicate that the constitutions capture helpful information about the preferences. Qualitatively, the generated constitutions (see Appendix H) often closely correspond to the principles described above.

## 4.2 HUMAN-ANNOTATED ALPACAEVAL DATA

We test our approach on human-annotated texts using the *AlpacaEval dataset* (Dubois et al., 2023). The dataset, used for the AlpacaEval leaderboard, features 648 data points cross-annotated by four annotators, with well-tested baseline AI annotators and evaluation tooling. The dataset captures general human preferences, likely very similar to the preferences language models used in this work have been fine-tuned on. As a consequence, the default annotator (without a constitution) agrees strongly with the annotations, leaving little room for improvement. The goal of this experiment, then, is not to exceed the default annotator's annotation performance on this dataset but to answer the following research questions: **(Q1)** Can constitutional annotators *match* the default annotator's performance on the aligned dataset, while providing the benefit of interpretable and editable constitutions? **(Q2)** Is ICAI able to extract and follow principles that are exactly opposite to the aligned dataset, despite the underlying model's biases? Note that most practical applications will be somewhere between the two extremes of the aligned and the unaligned scenario, allowing ICAI to increase the annotator's performance while offering insights into the learned principles, along with the ability to inspect and modify them as needed.

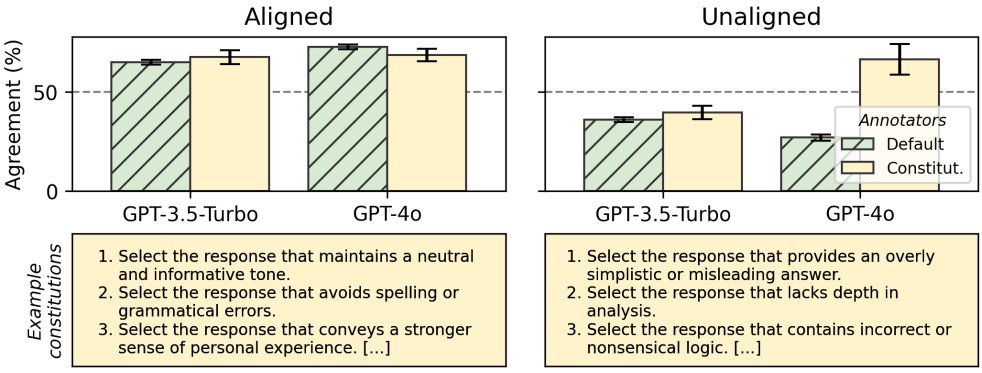

Figure 4: Results on AlpacaEval data. **GPT-4o generates and uses interpretable constitutions that match the performance of the default annotator on aligned preferences and notably increase agreement with unaligned preferences.** Tested on aligned (original) and unaligned (flipped) versions of AlpacaEval, with GPT-4o generating constitutions which are then used by constitutional annotators backed by GPT-4o and GPT-3.5-Turbo. Note we can only expect significant improvement in the unaligned case, as discussed in the main text. The aligned case does not leave room for improvement over the default annotator, but allows us to gain new insights into the preferences expressed in the dataset. In the unaligned case, GPT-4o's agreement improves notably, while GPT-3.5-Turbo's performance does not exceed random choice, indicating its limited ability to follow unaligned principles. Plots show mean and standard deviation (6 seeds).

**Experimental setup.** For each seed, we randomly select mutually exclusive training and test subsets with 65 annotated pairs each. Constitutions are generated on the training subset and results reported on the (unseen) test subset. We derive an *aligned* dataset based on majority vote (ties broken randomly) and an *unaligned* dataset using flipped annotations.

**Results.** Figure 4 shows that constitutional annotators approximately match default annotator performance in the aligned scenario while using an easily interpretable constitution, answering **(Q1)** affirmatively.[2] The unaligned dataset shows GPT-4o succeeding at following opposing principles, improving beyond the default annotator, while GPT-3.5-Turbo fails to do so despite using the same GPT-4o-generated constitutions. This answers **(Q2)** affirmatively for GPT-4o, revealing a capability gap between models. Since we evaluate the constitutions on an *unseen* test set, these results also demonstrate ICAI's potential for *annotation scaling*, extracting a constitution from a small training set (65 preferences here) and applying it to new data.

**Constitution transferability.** Given GPT-3.5-Turbo's limitations in the unaligned case in Figure 4, Appendix F.6 provides a more general exploration of how well our constitutions *can transfer between models*. To this end, we take the best performing constitution of the unaligned case on the training set and test how well models from Anthropic's Claude family are able to use these constitutions on the test set. The results indicate that this constitution transfers well to the Claude models — better than to GPT-3.5-Turbo, although transfer still incurs some loss. This is a promising result as it indicates that our constitutions are not excessively overfitting to the generation model, which indicates that they may capture more general concepts that are also interpretable to humans.

**Scaling.** Finally, although we consider sample-efficiency to be a benefit of our approach, we further evaluate whether these results also hold with larger scale datasets. We repeat the experiment on the AlpacaEval unaligned dataset with the full 648 preference pairs in the original dataset, using 324 samples each for training and testing. We observe that the overall results are very similar: the constitutional annotator with 61% agreement (66% prev.) still notably outperforms the the default annotator with 34% (27% prev.). The full results are included in Appendix F.7.

---

[2]We observe a very slight improvement in GPT-3.5-Turbo's annotations and a similarly small reduction in GPT-4o's agreement. This may be explained by the GPT-3.5-Turbo model being less attuned to the preferences in the dataset, which can be alleviated with a constitution. In the case of GPT-4o, however, the constitutional annotator likely focuses on the highly compressed constitution, even in cases where the default annotator would have a more nuanced (but less interpretable) understanding of the underlying preferences.

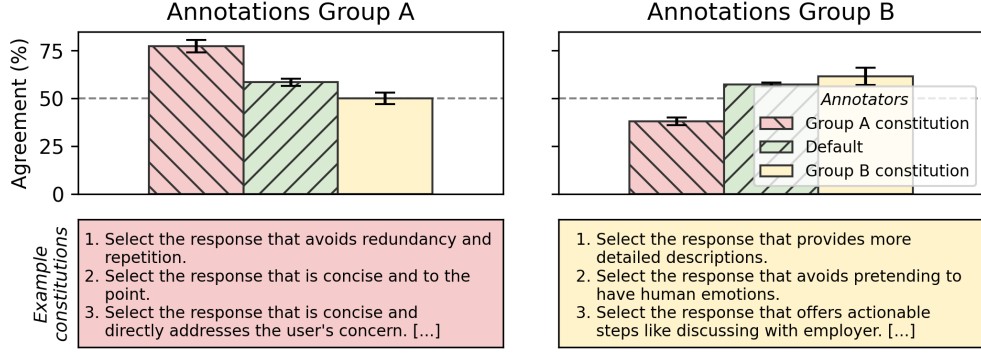

Figure 5: **Case-study: Constitutions for demographic groups on PRISM data.** We consider two groups reported by Kirk et al. (2024) to have preferences differing from average: participants born in one geographical region rank Mistral-7b higher in this dataset (*Group A*), and those born in another region rank Llama-2-7b lower than average (*Group B*). We generate constitutions for both groups to explore these preferences. For each group, the annotator using the group's data performs best. Constitutions (see Appendix H.4) suggest that Group A prefers Mistral-7b due to it's *conciseness*, while Group B's constitutions have recurring rules related to *providing more detailed descriptions*. Plots show mean and standard deviation (6 seeds) using GPT-4o.

### 4.3 INDIVIDUAL PREFERENCES: CHATBOT ARENA DATA

We evaluate ICAI's ability to generate *personal constitutions* using the *Chatbot Arena Conversations* dataset (Zheng et al., 2023), which consists of 33k human-annotated preferences with user-generated prompts, offering richer insights into individual preferences compared to AlpacaEval. We select two users exhibiting different preferences from the Chatbot Arena dataset (with 9 and 12 preference annotations respectively) and generate a separate constitution with 3 principles for each. Due to limited samples, there is no training-test split. We provide details on the experimental setup and results in Appendix F.1 and summarize the results here. As may be expected, we find that the personal constitutions generated for each user are able to improve the annotator's performance on the user's annotations, but do not transfer well to the other user's annotations. This shows that our constitutions successfully capture individual differences, illustrating the potential to generate personal constitutions. Note that how well a personal constitution transfers from one user to another depends on how similar their preferences are, with the contrast being the most pronounced between users with opposing preferences. Results will therefore vary for any given pair of users.

### 4.4 DEMOGRAPHIC GROUP PREFERENCES: PRISM DATA

Finally, we test ICAI's ability to help interpret demographic group preferences on the *PRISM* dataset (Kirk et al., 2024), consisting of annotations on 8,011 multi-turn conversations with 21 LLMs across a range of value-based and controversial topics from a diverse set of 1,396 annotators.

**Experimental setup.** We consider two annotator subgroups described by Kirk et al. (2024) to have differing preferences with respect to certain models relative to the average annotators. Group A, annotators born in the same geographical region, are reported to prefer the outputs of Mistral-7b relative to the overall rank. Similarly, Group B, annotators born in a different region, are reported to dislike Llama-2-7b disproportionately. These observations raise the question: *why do these subgroups prefer or dislike those specific models?* With no concrete explanation in the original paper, ICAI offers a method to generate and test possible explanations. We select the subset of PRISM interactions where Group A prefers Mistral-7b over another model (30 pairs[3]), and similarly, the subset where Group B rejects Llama-2-7b for any other model (80 pairs). Note that, as for Chatbot Arena, the few samples do not allow for a train/test split, which we consider acceptable for the purposes of data explanation. We then run ICAI on each subset separately to create a constitution for

---

[3]For each relevant PRISM datapoint, we pick one of three available rejected models at random.

Table 1: **Evaluation of possibly biased principles on three datasets.** Results on AlpacaEval (648 preferences), Chatbot Arena (5,115), and PRISM (7,490), showing relevance (fraction of data points where the principle applies) and accuracy (correctly reconstructed relevant data points). Grey values indicate relevance for fewer than 50 preferences. See Table 2 in Appendix F.2 for extended results.

| Principle 
 *Select the response that...* | AlpacaEv. | | ChatbotAr. | | PRISM | |
|---|---|---|---|---|---|---|
| | Acc | Rel | Acc | Rel | Acc | Rel |
| is overly lengthy and lacks brevity | 52.2 | 80.4 | 57.1 | 97.1 | 57.7 | 96.9 |
| provides a numbered list format | 73.4 | 12.2 | 62.3 | 45.5 | 71.6 | 17.7 |
| presents a definitive stance without nuance | 58.6 | 10.8 | 57.1 | 11.4 | 49.1 | 31.2 |
| is overly general and vague | 30.4 | 15.7 | 26.8 | 9.6 | 26.1 | 23.1 |
| emphasizes neutrality over providing information | 50.0 | 2.2 | 40.8 | 10.2 | 58.0 | 46.2 |

each group, testing each on both datasets. We use the same prompting setup as the Chatbot Arena experiments.

**Results.** Figure 5 shows that our constitutional annotators exceed default annotator performance in reconstructing the datasets they aim to compress but (as expected) do not transfer well to the other group's annotations, in line with the observation by Kirk et al. (2024) that each group's preference differs from average preferences. Indeed, the generated constitutions (see Appendix H.4) allow us to also ask *how* the preferences differ: Group A appears to strongly prefer more *concise* responses, whereas Group B has more diverse constitutions that often ask for *more detailed descriptions*.

## 4.5 APPLICATION: BIAS DETECTION

We showcase ICAI's application in bias detection, following three steps: (1) Generate and test 400 candidate principles on 1,000 preference pairs (500 from PRISM, 500 from Chatbot Arena[4]) to capture diverse biases (using ICAI algorithm steps 1 to 4). (2) Manually select principles indicating potential biases, focusing on those with high accuracy or limited applicability. (3) Evaluate these principles on 13k preferences (7,490 from PRISM, 5,115 from Chatbot Arena, and 648 from AlpacaEval), using step 4 of the ICAI algorithm. All steps use GPT-4o-mini for cost efficiency.

Results, shown in Table 1, reveal biases regarding verbosity, style, and assertiveness. Verbosity bias, where longer responses are preferred, is consistently observed across datasets, with Chatbot Arena and PRISM strongly favouring overly lengthy responses (notably, 57.1% and 57.7% accuracy, respectively). Style biases, such as a preference for numbered lists, are especially prominent in AlpacaEval (73% accuracy) and PRISM (72%), although their relevance is more limited compared to verbosity-related principles. Additionally, biases around ambiguity and vagueness vary by dataset; for example, PRISM annotations often favour neutral responses, while Chatbot Arena appears to actively select against response emphasizing neutrality. These results emphasize the dataset-specific nature of biases and demonstrate our framework's sensitivity to such patterns. More biases, discussion and mitigation strategies can be found in Appendix F.2. Further, we provide an additional application example of annotation scaling on helpful/harmless data in Appendix F.3.

## 4.6 ABLATION STUDIES

We conduct ablation studies to assess the contribution of each step in our pipeline across four scenarios: synthetic orthogonal, synthetic aligned, synthetic unaligned, and AlpacaEval unaligned, using GPT-3.5-Turbo for the first three and GPT-4 for the last. Numerical results, as well as a detailed discussion of the ablations, the experimental setup, and the results, are provided in Appendix F.4. Key findings are summarized below:

**Simplified principle generation (Step 1).** Generating only a single principle with a neutral prompt slightly reduces performance, indicating the importance of diverse principles. The performance drop is particularly pronounced on the synthetic aligned dataset, which aligns with our observation that GPT-3.5-Turbo struggles to generate both positive and negative principles from a single prompt.

---

[4]This experiment uses the Kaggle data (see Appendix A), differing from the data used in other experiments.

**No deduplication (Steps 2, 3, and 5).** Ablating deduplication produces mixed results: performance decreases on the synthetic aligned and synthetic unaligned datasets but improves on the synthetic orthogonal and AlpacaEval unaligned datasets. We hypothesize that repetition reinforces principles, especially when the model strongly opposes certain principles, as seen in AlpacaEval. Conversely, deduplication proves more effective when principles are less opposed to model biases, as observed in the synthetic orthogonal scenario. We conclude that deduplication is likely generally beneficial but that repetition may help overcome strong model biases in specific cases. Further details can be found in Appendix F.4.1.

**No filtering and testing (Steps 4 and 5).** Removing the filtering and testing steps has the most drastic impact, with performance dropping across all datasets. In particular, the annotators fail to outperform the random baseline in the unaligned experiments.

## 5 RELATED WORK

Our work focuses on deriving interpretable principles from human feedback data and using AI annotators to evaluate those principles. We build on work related to learning from human feedback, biases in feedback, interpretable preference models, and AI annotators.

**Learning from human feedback.** Fine-tuning LLMs with human feedback has significantly contributed to the success of modern LLMs (Ouyang et al., 2022; Stiennon et al., 2020). Typically, feedback is collected through pairwise comparisons of model outputs, training a *reward model* for fine-tuning, e.g. using reinforcement learning from human feedback (RLHF) (Ouyang et al., 2022; Stiennon et al., 2020; Kaufmann et al., 2024) or direct preference optimization (DPO) (Rafailov et al., 2023). Interpreting preference data is challenging since it generally lacks annotations of the underlying reasons and the reward model is often a black-box neural network, making it hard to interpret. Our work aims to generate interpretable principles explaining the feedback data.

**Biases in human feedback.** Identifying biases in human feedback data is crucial, as unintended biases are common. For example, Hosking et al. (2024) note a preference for assertiveness over truthfulness, while Wu & Aji (2023) highlight a bias towards grammatical correctness over factual accuracy. We discuss further prior work on annotator bias in Appendix B.2, also considering AI annotator bias. While these studies provide valuable insights, most methods for detecting biases rely on specialized feedback data collection, making them challenging to apply to pre-existing data. Our work generates interpretable principles from existing preference data, which can be inspected to detect biases and provide insights into the underlying preferences.

**LLM-based description of dataset differences.** Zhong et al. (2022; 2023) use LLMs to describe the difference between two text distributions in natural language, Dunlap et al. (2024b) a similar methodology for image distributions. Our work uses a related approach but adapted to the specific requirements and data structure of the pairwise preference domain.

**Interpretable preference models.** There has been a growing interest in creating interpretable preference models, aiding in understanding behaviour of AI systems. Go et al. (2024) create a *compositional* preference model based on 13 fixed features, similar to our constitutional principles. While they do not generate the constitution from data, they do create a regression model to weigh them, which would be a promising extension to our approach. Petridis et al. (2024) propose a feedback-based constitution generation method relying on interactive tools, whereas our approach can be directly applied to standard preference datasets.

**AI annotators.** Due to the cost and time required for human feedback collection, AI annotators, or LLM-as-a-judge, have been proposed as a scalable alternative. Constitutional AI (Bai et al., 2022b) uses LLMs with a set of principles for feedback. Human preference-based fine-tuning enables LLMs to generalize from very general principles, such as "do what's best for humanity" (Kundu et al., 2023), or even give feedback well-aligned with human preferences without any constitution (Zheng et al., 2023). Our experiments show similar trends, where default LLM annotators align well with dataset annotations, even without a constitution. AI annotators can also exhibit biases, further discussed in Appendix B. AlpacaEval (Li et al., 2024c) offers a set of well-validated AI annotators.

**Rule-based preference learning.** Rule learning, aiming to develop descriptive or predictive rules, has previously been applied to preference learning (de Sá et al., 2011). A common technique for rule learning is to first generate a set of candidate rules and then measure each rule's *support* in a dataset,

i.e. the fraction of data points that satisfy the rule (Fürnkranz et al., 2012; de Sá et al., 2011). Our algorithm follows this approach but, in contrast to more traditional rule learning, generates rules as natural language sentences. These rules, though more ambiguous and requiring AI annotators for interpretation, are expressive, interpretable, and easy for non-experts to edit[5]. Liu et al. (2023) follow a similar generate-and-test approach to derive text quality criteria. They require absolute scores on a fixed set of aspects, however, while our method leverages pairwise comparisons covering many aspects simultaneously, as is commonly the case for publicly available preference datasets.

**Concurrent work.** Further, we would like to highlight concurrent works by Kostolansky (2024), Kostolansky & Manyika (2024), Shankar et al. (2024b) and Dunlap et al. (2024a) exploring ideas highly related to our work, perhaps highlighting the timeliness of this line of research. Whilst related, our work differs in terms of the precise choice of approach taken as well as the comprehensiveness of our experiments. We provide a detailed comparison in Appendix B.3.

## 6 LIMITATIONS

When interpreting our results, there are several limitations to our approach important to consider. Firstly, *we do not show causality* — our generated principles correlate LLM annotators with the original annotations, but we cannot validate if these principles were used by the original annotators. Multiple constitutions may explain the data equally well (as in the Rashomon effect (Breiman, 2001)). Nonetheless, an undesirable principle correlating with annotations is concerning, even if the principle was not intentionally used. For example, in the "aligned" variant of the AlpacaEval dataset, some generated constitutions include principles to prefer verbose or redundant responses (see Appendix H.2.1). While this principle was likely not consciously followed by the original annotators, its high support in the dataset may warrant further investigation and possible data cleaning. We further discuss this limitation in Appendix C. Secondly, *constitutions represent a lossy compression* — a constitution of a few principles is a simplification of the decision-making process underlying annotations. Some annotations may not be possible to reconstruct based on a simple constitution. This trade-off highlights the tension between interpretability and accuracy: concise, human-readable principles versus more complex representations. While ICAI could be adapted for richer constitutions to balance this trade-off, a black-box reward model may be preferable when maximizing accuracy is critical. Finally, *preferences closely aligned to LLMs are challenging to test.* If an LLM annotator is already highly aligned with the dataset annotations, improving its performance with a constitution is challenging. The constitutional reconstruction loss is most useful for evaluating constitutions orthogonal to or against the popular opinions internalized by the LLM. On already well-aligned models, the constitution may not improve performance, but it can still provide insights into the underlying preferences. Future work should focus on addressing these limitations, extending the capabilities of our approach, possibly using multi-modal models, and exploring new applications.

## 7 CONCLUSION

We have presented our work on the *Inverse Constitutional AI* (ICAI) problem: first defining the ICAI problem compressing preference data into a short list of natural language principles (or *constitution*). We then introduced an initial ICAI algorithm as a first approach to generate such constitutions. We demonstrated the effectiveness of our approach in experiments across four different types of datasets: (a) *synthetic data* to provide a proof-of-concept; (b) *AlpacaEval data* to show the applicability to compress human-annotated data and the possibility of transferring constitutions across model families; (c) *Chatbot Arena data* to illustrate the generation of personal constitutions; and (d) *PRISM* data to demonstrate the ability to provide possible explanations for previously observed group preferences. We hope that our approach can improve both our understanding and the usefulness of widely-used feedback data. Potential use cases of our interpretable and editable constitutions and corresponding meta data include: highlighting issues with datasets, creating interpretable alternatives to black-box reward models, scaling human-annotated evaluation to new models and use cases, and improving model customization via personal or group constitutions. We look forward to future research that explores these use cases in detail.

---

[5]This can also be seen as a method for automatic prompt generation, as discussed in Appendix B.

ETHICS STATEMENT

We hope our work will have a positive societal impact by helping better understand preference data, already widely used for fine-tuning and evaluation of popular LLMs. We emphasize that our generated constitutions cannot claim to reconstruct an individual's true reasoning process. Similar to other interpretability methods, an ICAI constitution's principles may correlate with an individual's annotations but *no causal relationship* between the constitution and the annotator's reasoning can be proven. Further, our constitutions represent a notable compression of annotation considerations, which makes them highly interpretable but also means that they cannot reflect more multi-faceted decision making processes.

Thus, constitutions should be interpreted cautiously when working with human annotators to avoid potential negative implications. This is especially important when attempting to explain demographic preferences, as multiple possible explanations may correlate with the data and malicious actors could cherry-pick specific ones to make discriminatory statements or reinforce prior beliefs. Similarly, the use of our approach for personalized LLMs should also be considered carefully.

In general, we emphasize that our method can only provide information about specific *preference annotation datasets* rather than *annotators' reasoning processes* more broadly. To mitigate the potential for misinterpretation of results, we include a corresponding warning in our algorithm implementation that is shown to the user whenever a constitution is generated with ICAI.

When using ICAI for certain downstream use-cases, such as annotation scaling for training and evaluation or generating personal constitutions, there exists a risk of harmful bias amplification. If harmful biases exist in the original preference dataset, the ICAI constitution may pick up on these and propagate them downstream. This risk of amplifying biases is counterbalanced, however, by the ability to edit and inspect the generated principles. This ability potentially helps avoid amplification of unintended biases when using ICAI in downstream applications. This potential visibility of biases in ICAI distinguishes our method from other widely-used methods using preference data, such as black-box reward models or aggregate evaluation statistics, which make such biases more difficult to detect. We recommend users of our method to always take a close look if generated constitutions are aligned with their own values and contain any potentially harmful biases before proceeding with downstream use-cases. Overall, we believe the potential for positive impacts outweighs possible negative impacts.

REPRODUCIBILITY STATEMENT

We make the source code for our method and experiments publicly available at `https://github.com/rdnfn/icai`. Further, we attempted to add as many details in the paper as possible, including all prompts in Appendix G as well as a description of our synthetic data generation approach in Appendix I. For direct comparability, we also make numerical results available in Appendix F.9.

ACKNOWLEDGEMENTS

We thank Benjamin Minixhofer, Max Bartolo, Tom Hosking, Hritik Bansal, the RECOG-AI team at the Leverhulme Centre for the Future of Intelligence (especially José Hernández-Orallo and Lorenzo Pacchiardi), Bill Marino and Jason Brown for their valuable feedback and discussions on early versions of this work. Further, we thank all reviewers for taking the time to read our work and to provide helpful feedback. We also thank Anthropic for providing free research access to their models. Arduin Findeis was supported by a University of Cambridge School of Physical Sciences Award and by the UKRI Centre for Doctoral Training in Application of Artificial Intelligence to the study of Environmental Risks (reference EP/S022961/1). This publication was further supported by LMUexcellent, funded by the Federal Ministry of Education and Research (BMBF) and the Free State of Bavaria under the Excellence Strategy of the Federal Government and the Länder as well as by the Hightech Agenda Bavaria.

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

APPENDIX

# A  DATASET DETAILS

We use four datasets in our experiments: synthetic data, AlpacaEval, Chatbot Arena, and PRISM. The synthetic dataset is described in detail including the generation process in Appendix I, the other datasets are publicly available and described in the following.

**AlpacaEval** is a dataset of 648 human-annotated preferences, each consisting of a pair of model outputs, with one preferred over the other. It is licensed under CC-BY-NC-4.0 and can be accessed at `https://huggingface.co/datasets/tatsu-lab/alpaca_eval`. To reduce computational costs and highlight the sample efficiency of our approach, we typically use a subset of 130 datapoints (65 train, 65 test). For larger-scale evaluation in Appendix F.7, we use the full dataset of 648 datapoints (324 train, 324 test), which we refer to as AlpacaEval-Large.

**Chatbot Arena Conversations** is a dataset of 33,000 preferences from the Chatbot Arena, used by the popular LMSYS leaderboard. Each datapoint consists of a prompt and preference over a pair of model outputs, both human genearted. It is licensed under CC-BY-NC-4.0 and can be accessed at `https://huggingface.co/datasets/lmsys/chatbot_arena_conversations`.

**Chatbot Arena Kaggle** is a dataset of 55,000 human-annotated preferences, each consisting of a pair of model outputs, with one preferred over the other. The dataset is similar to Chatbot Arena Conversations, but contains more recent data. It is licensed under CC BY-NC 4.0 and can be accessed at `https://www.kaggle.com/competitions/lmsys-chatbot-arena/data`.

**PRISM** is a dataset of 8,011 human-annotated preferences, each consisting of a pair of model outputs, with one preferred over the other. The dataset is licensed under CC-BY-4.0 and can be accessed at `https://huggingface.co/datasets/HannahRoseKirk/prism-alignment`.

**Anthropic HH-RLHF** is a collection of human-annotated preference datasets by Bai et al. (2022a) focused on annotations preferring helpful and harmless outputs, with approx. 44,000 and 42,000 conversations respectively. The helpfulness data, similar to other datasets, contains general model use-cases, with the more helpful of two responses selected. The harmless dataset is based on red-teaming prompts that explicitly aim to elicit harmful responses from models. The less harmful response is selected. The data is available under MIT license at `https://github.com/anthropics/hh-rlhf`.

# B  EXTENDED RELATED AND CONCURRENT WORK

In addition to the related work discussed in the main body (Section 5), in a broader sense ICAI can also be viewed as a method for automated prompt generation. Another relevant area concerns biases exhibited by AI annotators, which are important due to ICAI's reliance on such annotators and its potential use as a tool for detecting these biases. We also give a more extensive discussion of concurrent work, in addition to the brief overview in the main body (Appendix B.3).

## B.1  RELATION TO PROMPT GENERATION

LLM outputs can be guided by generating specific prompts. This relates closely to our work, where we create principles to steer outputs.

Manual adversarial prompt generation, or 'jailbreaking', allows users to bypass safety constraints imposed during fine-tuning. This process can also be automated (Zou et al., 2023), generating adversarial prompts to attack a wide range of models. Li et al. (2024a) propose virtual tokens to steer outputs towards specific viewpoints, using a dataset of question responses to define these personas, unlike our approach based on pairwise comparisons and interpretable constitutions. Rodriguez et al. (2024) explore the use of LLMs to discover and classify user intent, which may help adapt model prompts.

## B.2 RELATION TO ANNOTATOR BIASES

Bansal et al. (2024) highlight that feedback methods influence biases, e.g., annotators focus more on accuracy in pairwise comparisons compared to rating feedback. Additionally, Sharma et al. (2023) observe a bias towards sycophantic outputs, where responses align with the user's beliefs rather than the truth.

Further, AI annotators can exhibit biases, partially overlapping with human biases (Chen et al., 2024), and inconsistencies in their judgements (Stureborg et al., 2024). Examples include position bias (preferring the first output) (Zheng et al., 2023), verbosity bias (preferring longer responses) (Zheng et al., 2023), and self-enhancement or familiarity bias (preferring outputs similar to their own) (Panickssery et al., 2024; Stureborg et al., 2024). Their proposed mitigation measures include trying both orderings and tying if inconsistent, with further explorations in later work (Dubois et al., 2024).

## B.3 EXTENDED CONCURRENT WORK

**Inverse Constitutional AI.** Kostolansky (2024) introduced (and identically named) the problem of *Inverse Constitutional AI* (ICAI), concurrently with our work. Their problem formulation includes our first step, going from preferences to principles, but omits the reconstruction loss using a constitutional annotator. Their corresponding method also differs from ours, first clustering principles and then generating principles per cluster (instead of the other way around). Their results focus on reconstructing known clusters of preferences — requiring special preference datasets with known clusters and providing limited insight into the usefulness of each cluster's generated principles. Thus, their results cannot be directly compared to ours. Based on our ablation results, we remain sceptical that more focus on clustering would be helpful to create representative principles.

**Iterative Inverse Constitutional AI (I³CAI).** Kostolansky & Manyika (2024) also concurrently introduced an alternative formulation named *Iterative Inverse Constitutional AI* (I³CAI). This problem formulation focuses on optimising each principle's ability to nudge an LLM towards correctly reconstructing annotations. This objective resembles our per-principle voting step, although being based on the conditional probability of correctly annotating a pair given a principle — rather than observed sampled annotations. This approach may offer a less noisy estimate than sampling but is not possible for all non-open API-based models. Perhaps due to this limitation their experiments use the Llama-2-7b model, relatively weak compared to larger state-of-the-art models. As the name suggests, their method *iteratively* refines principles, differing quite a bit from our approach and requiring a "seed" constitution to initialize the process. As their implementation is at the time of writing not publicly available (as far as we are aware), we were unable to directly evaluate their method relative to ours — but testing this method in our experimental settings would be an interesting future study to run.

**SPADE.** Shankar et al. (2024b) adapt *System for Prompt Analysis and Delta-Based Evaluation* (SPADE) (Shankar et al., 2024a) to the pairwise annotation setting. SPADE is a method that was originally designed for generating evaluation criteria based on differences ("deltas") between different prompt versions during the development of LLM-based applications. The authors adapt this method to the pairwise setting and report it as a baseline, but limited information regarding the transfer of SPADE to the pairwise setting makes it challenging for us to make a meaningful comparison. Based on the original SPADE paper, this method likely uses a two stage process: (1) initially proposing criteria based on the preference data (prompt deltas originally) using an LLM, and then (2) testing each criterion and selecting a subset based on its coverage of test cases as well as false failure rate. We believe that adding a similar selection process of rules, whilst adding complexity, would be an interesting extension of our method to explore in future work. Our initial principle selection process is intentionally simpler to avoid introducing more complexity. Regarding larger datasets and associated costs, we are uncertain whether and how they adapt their method to scale and whether they add any form of clustering. We were unable to find a public implementation of this pairwise SPADE version.

**VibeCheck.** Dunlap et al. (2024a) propose *VibeCheck*, a system that automatically detects qualitative differences ("vibe axes") between models based on various datasets, including pairwise preferences. The corresponding algorithm follows a similar overall approach as ours: initially proposing vibes (similar to our principles) using LLMs and the validating the generated vibes using LLM-

as-a-Judge systems in a filtering step. Different to our work, vibes are phrased as qualitative axes (e.g. from formal to friendly language) rather than principles explicitly instructing an LLM. Further, VibeCheck's algorithmic steps appear to primarily optimise towards vibes that are well-defined (consistent across annotators) and enable effective separation between models, rather explicitly optimising for the ability to accurately reconstruct preferences as well as possible. Whilst not explicitly optimising for annotation reconstruction during the vibe generation and selection, the authors do create a preference prediction model using a logistic regression on top of the vibe features — a model with a use-case comparable to our constitutional annotators. The experimental results for VibeCheck similarly focus on model separation and do not include LLM-as-a-Judge or reward model baselines as our work does. Thus, there are no directly comparable results of the logistic regression model to our approach. Overall, VibeCheck represents a promising approach that would be interesting to apply directly to our Inverse Constitutional AI problem of reconstructing preferences, possibly adjusting the algorithm's internal optimisation objective more directly towards preference reconstruction. Notable algorithmic aspects from VibeCheck that would be potentially interesting to integrate in future ICAI algorithm versions would be the iterative refinement of vibes, self-consistency checks, and the multi-judge setup. It is worth noting that our ICAI method can also be used to quantify differences in model behaviour: using the same setup as in our bias detection example (Section 4.5) on data subsets where individual models win, we can measure models qualities based on our generated principles. We would welcome future work that adapts VibeCheck to provide a more direct comparison to ICAI, and vice versa.

**PopAlign.** Wang et al. (2024) introduce PopAlign, a method that aims to improve model alignment and robustness by synthesizing more diverse response pairs as well as the corresponding preference data. Among the proposed diversification strategies, the elicitive contrast approach is particularly related to our work, as it prompts the model to first derive principles for a given instruction (based on overarching 'helpful and harmless' guidelines) and then use these principles to generate a response. While this dynamic principle derivation resembles our approach, PopAlign focuses on generating new feedback data, whereas ICAI aims to interpret existing datasets. As a result, PopAlign does not incorporate responses and preferences into its principle generation process, nor does it aim to produce globally applicable principles evaluated across other data points. While the goals of the two methods differ, they are complementary: ICAI's data-driven constitutions could inform the creation of more targeted contrastive prompts for PopAlign, while PopAlign's strategies for generating diverse responses could provide richer preference data for ICAI, enabling a more comprehensive understanding of human preferences.

**Rule Based Rewards (RBR).** Mu et al. (2024) introduce a method to train auxiliary 'rule-based reward models' by composing natural-language 'propositions' — binary statements about a response, such as "contains an apology" — into a linear combination that serves as a reward signal. These propositions are hand-crafted and used as features in the reward model[6], rather than as direct preference indicators. While related, this approach does not aim to interpret or compress existing feedback datasets. Mu et al. (2024) further emphasizes detailed, interpretable rules over broader principles (e.g., "prefer the helpful response") to improve interpretability and steerability An ICAI-like approach could complement rule-based methods by generating candidate propositions in a data-driven manner, potentially reducing manual engineering effort and enabling efficient fine-tuning of models with modified constitutions. This highlights the potential for combining principled compression with explicit, rule-based rewards to enhance both interpretability and adaptability in safety-critical applications.

## C   FURTHER LIMITATION DISCUSSION

**Non-uniqueness and variability of constitutions.** An important limitation of our method is that a well-performing constitution is rarely *unique*: annotators with multiple potentially quite different constitutions may achieve equivalent performance across a dataset. Breiman (2001) more generally describes this non-uniqueness as the *Rashomon* effect, applying to many machine learning problems. In the context of an interpretability framework, such as ours, this effect needs to be carefully considered whenever drawing conclusions. If an individual principle or constitution is able to help reconstruct a certain subset of annotation well, there may be many other principles or constitutions

---

[6]They can also be used as atoms in hand-crafted rules.

that reconstruct. Thus, we cannot claim any *causal relationship*: a well-performing principle *does not mean* that the annotator (AI or human) used that principle to create the annotations.

Nevertheless, in the context of bias detection, knowing that a harmful principle works well to reconstruct a specific dataset can still be very useful. Even if we cannot know if the annotator willingly used such a principle, we know the data *can be interpreted* to encode this harmful principle. Downstream applications (e.g., reward models) may make the same interpretation of the original dataset, and encode such a principle (possibly in a way that is hard to detect such as millions of model parameters). Our method has the potential to highlight such potential harmful principles, enabling preference data users to mitigate these biases, for example by data filtering or collecting additional data.

On the other hand, the non-uniqueness of constitutions and principles means that our method is *very unlikely to find all potential harmful biases* in the context of bias detection. It is therefore critical to avoid interpreting the lack of harmful biases in our constitutions as an indication that no harmful biases are present in a given dataset. Given the diversity of potential harmful biases in commonly used datasets, our method should not be misunderstood to be able to find *all* harmful biases. However, the more prominent a bias is, the more likely it is that a corresponding principle is reliably generated and promoted to the constitution. Thus, ICAI serves as a valuable tool for detecting many — but not all — biases in preference datasets.

In the context of annotation scaling, well-performing constitutions can provide an effective way to scale small-scale human annotations to larger datasets. However, our constitution, like the parameters of alternative methods of annotation scaling (e.g. the PairRM baseline by Jiang et al. (2023) or LLM-as-a-Judge (Zheng et al., 2023)), is not unique and there may be many different alternative models that achieve equivalent performance. A benefit with our approach is that these differences are more transparent and interpretable: It is more challenging to tell what the difference is between two sets of reward model parameters than between two constitutions.

## D    INCLUDED MODELS

Throughout our experiments, we primarily use the following three models:    OpenAI's `gpt-3.5-turbo-0125` (referred to as *GPT-3.5-Turbo*), `gpt-4o-2024-05-13` (referred to as *GPT-4o*) and `text-embedding-ada-002` embedding model for clustering steps in the algorithm (across all experiments). Detailed model descriptions of these OpenAI models are available at `https://platform.openai.com/docs/models/`. Certain experiments use additional models, these are described in the relevant experiments discussions.

## E    COST ESTIMATES

In this section, we estimate the cost of reproducing the main experiments shown in this paper. All experiments were run using models via API access from OpenAI and Anthropic. Note that all estimates are subject to variability due to provider pricing as well as inherent variability of the length (and thus cost) of model outputs.

*Note that this cost estimate excludes the scale-up, ablation and PRISM experiments.*

**Synthetic experiments.** The first set of experiments are the synthetic experiments, which are entirely run using GPT-3.5-Turbo. Per run (30 samples, 1 constitution, annotation on same 30 samples) these experiments cost approximately 0.05$. Overall, we estimate it would cost 2.7$ to re-run all experiments shown (3 datasets × 6 seeds).

**AlpacaEval experiments.** The second set of experiments are the AlpacaEval experiments, split into the main aligned/unaligned experiments as well as cross-model experiments. The main experiments cost approx. 2.20$ per seed. Overall, we estimate it would cost 26.40$ to re-run all of the main experiments (2 datasets × 6 seeds). Additionally, we estimate the cross-model (just annotation) experiments would cost 5.00$.

**Chatbot Arena experiments.** The third set of experiments are the Chatbot Arena experiments, split into the main aligned/unaligned experiments as well as cross-model experiments. The main

experiments cost approx 1.10$ per seed. Overall, we estimate it would cost 13.20$ to re-run all of the main experiments (2 datasets × 6 seeds).

We estimate the remaining cost of experiments to be less than 5$. Overall, we thus estimate the total cost of re-running our experiments to approx. 52.30$ in API costs. Note that the overall cost for running experiments in the context of this project was about 3 times this amount (approx. 156.90$), due to failed runs and additional experimentation that did not fit into the scope of the paper.

## F    ADDITIONAL EXPERIMENT DETAILS

In this section, we provide additional details and results for experiments discussed in the main text.

### F.1    DETAILED CHATBOT ARENA EXPERIMENTS

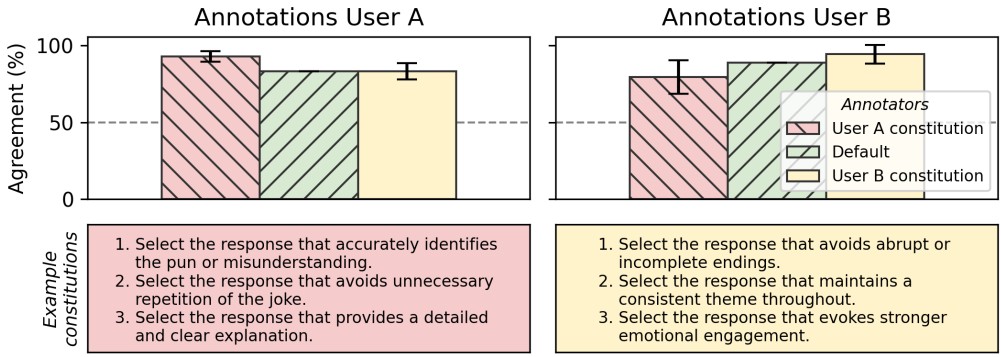

Figure 6: **Case-study: Personal constitutions for anonymous Chatbot Arena users.** Personal constitutions have the potential to help make LLM applications more helpful and customized to individual users' preferences — in an interpretable way. We generate constitutions based on a single user's annotations and check the constitutions' ability to help reconstruct the annotations of the same user and another user. For the two users, selected to have differing preferences, we observe that generated personal constitutions appear to work best for the original user and not transfer perfectly to another user. Note that this effect will vary for other users depending on how different users' preferences are. Plots show mean and standard deviation (6 seeds) using GPT-4o.

**Experimental setup.** We select two users exhibiting different preferences[7] from the Chatbot Arena dataset (with 9 and 12 preference annotations respectively) and generate a separate constitution with 3 principles for each. Note that due to the small number of samples, there is no split between training and test data. While this lack of separation may result in overfitting, we consider it acceptable in this case since our goal is not to develop a generalizable model, but rather to explain the preferences within this specific dataset. To better detect the effect of user-specific principles, we adapt our generation and annotation prompts (Appendix G) to generate constitutions more specific to the individual users and follow the specific principles more closely rather than the model's priors.

**Results.** The results can be found in Figure 6 and are discussed in the main text (Section 4.3).

### F.2    USE-CASE EXAMPLE: BIAS DETECTION

LLMs are known to exhibit biases, often originating from the data used to train them. This includes stylistic biases (Dubois et al., 2023; 2024), social or stereotypical biases (Navigli et al., 2023), and cultural biases (Tao et al., 2024). They can arise from the initial training data (Navigli et al., 2023) as well as the feedback data used during fine-tuning (Dubois et al., 2024; Tao et al., 2024). Our framework, ICAI, provides a mechanism to detect such biases in the preference data and offers actionable tools to analyse and mitigate them. This section presents examples of biases identified by

---

[7]The Chatbot Arena dataset was filtered by the authors to avoid personally identifiable information (PII).

Table 2: **Evaluation of possibly biased principles on three datasets.** Results on AlpacaEval (648 preferences), Chatbot Arena (5,115), and PRISM (7,490). Metrics shown are relevance (fraction of data points where the principle applies) and accuracy (fraction of relevant data points correctly reconstructed). Gray values indicate relevance for less than 50 preferences on the dataset. Hand-picked, grouped and sorted to illustrate presence of well-known and less discussed biases.

| Principle _Select the response that..._ | AlpacaEv. Acc | Rel | ChatbotAr. Acc | Rel | PRISM Acc | Rel |
|---|---|---|---|---|---|---|
| _Verbosity and style_ | | | | | | |
| is overly lengthy and lacks brevity | 52.2 | 80.4 | 57.1 | 97.1 | 57.7 | 96.9 |
| contains redundant information | 44.7 | 7.3 | 39.7 | 12.8 | 35.1 | 3.0 |
| provides a numbered list format | 73.4 | 12.2 | 62.3 | 45.5 | 71.6 | 17.7 |
| is more concise and structured | 47.4 | 98.6 | 42.9 | 95.5 | 42.7 | 94.6 |
| uses more formal language | 56.8 | 6.8 | 53.4 | 16.0 | 61.9 | 5.4 |
| feels more casual and friendly | 43.8 | 67.3 | 43.3 | 61.2 | 41.1 | 74.5 |
| _Assertiveness_ | | | | | | |
| presents a definitive stance without nuance | 58.6 | 10.8 | 57.1 | 11.4 | 49.1 | 31.2 |
| presents a biased viewpoint without nuance | 36.8 | 2.9 | 50.8 | 4.6 | 45.4 | 17.2 |
| lacks nuance in political analysis | 36.4 | 1.7 | 49.3 | 2.7 | 44.0 | 11.1 |
| lacks neutrality in political matters | 55.6 | 1.4 | 49.6 | 2.7 | 38.9 | 9.8 |
| presents a one-sided argument | 32.0 | 3.9 | 53.4 | 5.2 | 46.1 | 19.5 |
| avoids acknowledging complexity of the issue | 28.1 | 4.9 | 37.9 | 10.2 | 36.9 | 25.8 |
| promotes divisive political statements | 50.0 | 0.9 | 52.2 | 1.8 | 39.2 | 6.6 |
| assigns sole blame without context | 80.0 | 0.8 | 53.9 | 1.5 | 41.4 | 6.2 |
| does not consider personal preferences | 100.0 | 0.5 | 42.4 | 1.7 | 50.5 | 6.8 |
| _Ambiguity and vagueness_ | | | | | | |
| is overly general and vague | 30.4 | 15.7 | 26.8 | 9.6 | 26.1 | 23.1 |
| emphasizes neutrality over providing information | 50.0 | 2.2 | 40.8 | 10.2 | 58.0 | 46.2 |
| presents ambiguous or non-committal language | 100.0 | 0.2 | 30.4 | 1.3 | 46.8 | 13.0 |
| introduces ambiguity about the assistant's nature | 100.0 | 0.2 | 33.0 | 3.5 | 44.0 | 14.0 |

our approach in the AlpacaEval and Chatbot Arena datasets and further discusses possible mitigation strategies and limitations.

**Bias detection study.** Our method can help detect biases by generating principles that expose the bias and measuring their performance. We run a study to showcase this ability of ICAI using the following procedure: (1) We generate a set of candidate biases (principles) using ICAI on a training set of 1,000 preference pairs, 500 from PRISM and 500 from Chatbot Arena. Note that we use the newer Chatbot Arena Kaggle dataset for this study (see Appendix A), differing from the main experiments. To ensure a diverse set of principles, including potentially problematic ones that may only apply to a small subset of the data, we generate and test 400 principles (number of clusters) on this initial subset. (2) We manually select a subset of principles that we consider to be potential "problematic" biases. We focus on biases that perform well in terms of accuracy on the initial test set but also consider biases that are non-relevant for the vast majority of the training set — and thus have less reliable accuracy measurement (as that is based on the number of relevant data points). (3) We then re-run the principle testing step of our pipeline on a much larger test set of over 13k preference pairs, consisting of 7,490 preferences from PRISM, 5,115 from Chatbot Arena, and the entire dataset of 648 cross-annotated AlpacaEval preferences. To run such a large study cost-effectively, each component of our algorithm uses GPT-4o-mini (rather than GPT-4o). The results are shown in Table 2.

**Verbosity bias.** One of the most well-known biases in preference data is verbosity bias, where longer responses are preferred. While both humans and AI annotators exhibit this bias (Dubois et al., 2023; Chen et al., 2024), AI annotators seem to place excessive focus on this trait at the cost of other important aspects, leading to problematic artefacts in evaluation and training of language models (Dubois et al., 2024). We observe strong bias towards longer responses on both Chatbot Arena and

PRISM[8], with principles preferring responses that are *"overly lengthy and [lack] brevity"* achieving notably above random accuracy (acc. of 57.1 and 57.7% respectively) on a significant portion of the dataset (rel. of 97.1 and 96.9% respectively). Note that the bias is less pronounced on the AlpacaEval dataset in our experiments (acc. of 52.0% and rel. of 80.4%), despite prior work noting a preference for longer responses in that dataset (Dubois et al., 2024). Even though apparently less pronounced than in other datasets, the AlpacaEval verbosity bias is also reflected in the constitutions generated for the aligned AlpacaEval dataset (see Appendix H.2.1 for a sample), where principles favouring verbose or redundant responses appear in 3 out of 6 seeds. The PRISM and Chatbot Arena experiments in the main paper focus on specific subsets of the dataset, however, and the constitutions generated for those subsets are not directly comparable to the data in Table 2. For instance, contrary to the general trend visible in Table 2, the constitutions generated for Group A of the PRISM dataset seem to favour conciseness over redundancy, while Group B's constitutions are more in line with the general trend (see Appendix H.4). This outcome further validates our framework's capability to detect biases that are dataset-specific.

**List bias.** A similarly well-known bias is the preference for structured responses, Markdown syntax, and lists in particular (Li et al., 2024b). We see this style bias reflected in Table 2, with the rule favouring *"the response that provides a numbered list format"* achieving high accuracy across all datasets, especially AlpacaEval (73%) and PRISM (72%), although this principle is far less broadly applicable than the ones centred on verbosity (rel. of 12.2 and 17.7% respectively).

**Assertiveness bias.** Finally, we observe a preference for **assertiveness** (as discussed by Hosking et al. (2024)), with principles favouring *"the response that presents a definite stance without nuance"* and similar (see Table 2) performing well on AlpacaEval and Chatbot Arena. This bias is notably common in political contexts (compare Table 2), giving cause for concern. It is quite possible, however, that preferences supporting this bias were given to counteract the language model's tendency to over-qualify or hedge its statements (related to the next paragraph), so it is important to consider the context in which this bias is observed.

**Ambiguity or vagueness.** In addition to these well-known verbosity and style biases, we also observe less extensively discussed biases, commonly centring around ambiguity and vagueness: the principle *"Select the response that emphasizes neutrality over providing information"* performs well on PRISM but has lower than random accuracy on Chatbot Arena data, indicating this rule is not followed, on average, by Chatbot Arena annotations. Neutrality in the response appears to be appears to be actively selected *against*, on average, in the Chatbot Arena subset. The Chatbot Arena annotations further do not appear to, on average, reject responses that *"promote divisive political statements"*, unlike PRISM annotations. Further, we observe that Chatbot Arena annotations actively select against responses that *"acknowledge limitation in available information"*.

**Mitigation.** The results above indicate that ICAI is able to both find well-described and less widely discussed biases in pairwise preference data. Once biases are identified, ICAI offers actionable strategies for mitigation. Possible avenues include (1) synthesizing new preferences with a modified constitution that avoids the bias, and (2) curating the training dataset by filtering or balancing preferences to reduce bias.

The second approach leverages ICAI's ability to measure the support of each principle within the dataset, identifying which preferences align with the bias. By removing or rebalancing these preferences, biases can potentially be mitigated. This approach also allows for a more detailed analysis of the data, making it possible to develop tailored strategies, such as refining the preference collection process. Going beyond removal, our framework can also be used to identify and promote preference pairs that counteract the bias, i.e., agree with an opposing principle. For example, despite the verbosity bias in AlpacaEval, one generated constitution includes a principle favouring concise responses, indicating that the dataset contains a substantial portion of counteracting preferences. ICAI thus serves as a versatile tool for dataset filtering, balancing, and curation, which have proven effective in other contexts (Liu et al., 2024; Park et al., 2024). We are excited for future work to explore these mitigation strategies in more detail.

**Limitations.** Our methods' ability to detect biases depends on two factors: the diversity of candidate principles and reliability of the filtering mechanism. While stylistic biases, such as verbosity and list

---

[8]This is despite PRISM attempting to alleviate verbosity bias by instructing the LLM to produce shorter responses (Kirk et al., 2024).

preferences, are straightforward to detect, social and cultural biases can be more challenging, since they are often expressed in subtle ways. These biases, including those related to gender or minority representation, are critical to address. However, in the datasets analysed, our constitutions do not show direct evidence of such biases, likely due to the limited dataset size and the constitutions' focus on broadly applicable principles. Unlike stylistic biases, social and cultural biases often affect smaller subsets of data and may coincide with alternative explanations for preferences.

Detecting these subtler biases requires expanding the dataset, increasing the number of candidate principles generated per preference, and increasing the scope of the analysis beyond the top principles to those that, while not universally applicable, exhibit strong predictive power for specific data subsets. A thorough investigation of social and cultural biases using ICAI represents a promising direction for future research.

### F.3    Use-case Example: Annotation Scaling on Helpful/Harmless Data

Collecting human annotations for specific purposes can be expensive and time-consuming. We demonstrate the use of ICAI to scale up preference annotations $10\times$ based on a small set of 100 initial ground-truth annotations to 1000 new response pairs. In particular, we consider the use of ICAI to scale up *harmlessness* and *helpfulness* annotations, using the *Anthropic HH-RLHF* dataset by Bai et al. (2022a). More information about the dataset is available in Appendix A.

**Experimental setup.** We randomly sample two *training sets* of 100 data points each from separate helpful and harmless datasets in *Anthropic HH-RLHF*.[9] The *helpful* and *harmless* datasets contain human annotations that prefer more helpful and harmless responses, respectively. We similarly sample two separate *test sets* of 1,000 data points from each dataset. We then apply ICAI on each training set to create two separate constitutions, one harmless and one helpful, and test the ability of an LLM to use these constitutions to reconstruct each dataset. We use GPT-4o-mini (`gpt-4o-mini-2024-07-18`) for all parts of the ICAI algorithm, and the constitutional and default annotations. We slightly adjust the principle proposal and voting prompts in the ICAI algorithm to accommodate the long multi-turn nature of the Anthropic HH preference dataset.[10]

**Results.** The results are shown and discussed in Figure 7. Results shown are mean and standard deviation over 3 seeds of the entire pipeline.

### F.4    Ablation details

We provide detailed numerical results for and discussions of the ablation experiments introduced in Section 4.6. Each experiment is averaged over six seeds, with annotator agreement and confidence intervals shown in Table 3. Below are the specifics of each ablation:

**Simplified principle generation (Step 1).** In this ablation, we generate principles using a single neutral prompt instead of multiple prompts. As shown in Table 3, this leads to a reduction in annotator agreement across all datasets, with the largest drop in the synthetic unaligned dataset. This confirms our hypothesis that GPT-3.5-Turbo struggles with generating both positive and negative principles from a single prompt.

**Principle generation with multiple preferences (Step 1)** We test the effect of prompting with multiple preferences simultaneously to generate the principles in Step 1. By default, only a single preference is used in the prompt. In these experiments, we give the model 5 preferences simultaneously, and then ask the model to generate 10 corresponding principles. We randomly group all preferences into groups of size 5, that are then used to prompt the model in Step 1. We observe mixed results: for some scenarios (Synth aligned and AlpacaEval unaligned) the model improves performance whereas for the others this configuration decreases performance. To maximize principle diversity, it may be useful to combine both single and multi-preference principle generation.

---

[9]All data is sampled from the `train.jsonl.gz` files in the helpful-base/harmless-base subdirectories of the data repository.

[10]In particular, we add additional separators between the responses ("`---`") and explicitly prompt the model to focus on the last conversation turn.

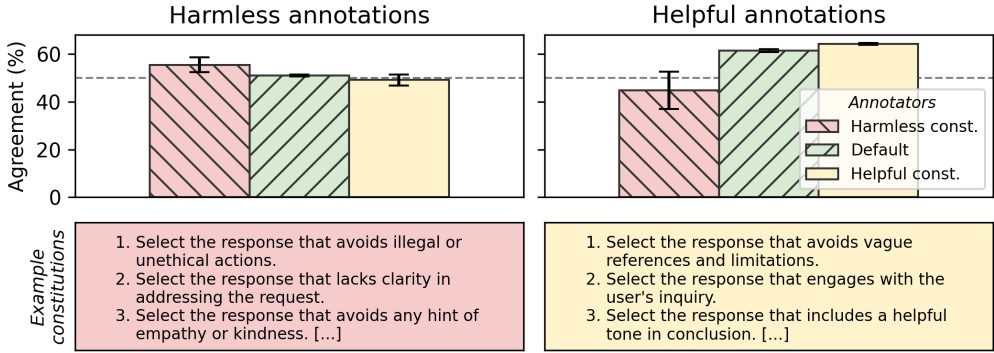

Figure 7: **Use-case example: scaling annotations 10× with ICAI on helpful/harmless preference data.** We observe that our constitutional annotators are able to outperform the default baseline annotator on each dataset. Qualitatively, each dataset's responses are clearly distinguishable from each other. For example, all harmless constitutions contain principles to avoid promoting illegal actions whilst the helpful ones often focus on helpful tone and user engagement. Quantitatively, we annotators with harmless constitutions do not appear to transfer well to helpful data and vice versa. Our experiments closely replicate findings by Bai et al. (2022a), indicating that these two datasets encode anti-correlated objectives: a fully harmless response should refuse to be helpful for harmful actions.

Table 3: Results for ablation of different pipeline components. The table shows the mean agreement and standard deviation over 6 seeds. Each configuration is tested over four different datasets from Section 4, based on the synthetic (*Synth*) and AlpacaEval (*AE*) datasets. The best result per row is highlighted in bold.

| Name | Original | Single Princ. (S1) | Multi-Pref. (S1) | No De-Dup. (S2 & S3+) | No Test/Filter (S4 & S5) |
|---|---|---|---|---|---|
| Synth Orth *(GPT-3.5-Turbo)* | **86.7** ± 8.4 | 82.2 ± 4.6 | 83.3 ± 11.2 | 80.0 ± 17.0 | 51.7 ± 13.6 |
| Synth Aligned *(GPT-3.5-Turbo)* | 92.2 ± 6.9 | 89.4 ± 11.6 | **98.3** ± 4.1 | 93.9 ± 8.8 | 69.4 ± 30.9 |
| Synth Unaligned *(GPT-3.5-Turbo)* | **84.4** ± 9.8 | 62.8 ± 31.0 | 69.4 ± 19.8 | 83.9 ± 8.3 | 31.1 ± 18.3 |
| AE Unaligned *(GPT-4o)* | 66.4 ± 7.7 | 65.9 ± 2.8 | **72.1** ± 2.0 | 70.0 ± 2.9 | 40.8 ± 11.3 |

**No deduplication (Steps 2, 3, and 5).** We ablate deduplication by testing all generated principles without removing duplicates. The results, shown in Table 3, are mixed: performance decreases on the synthetic aligned and synthetic unaligned datasets but improves on the synthetic orthogonal and AlpacaEval unaligned datasets. This suggests that repetition may help reinforce principles in datasets where the model holds strong prior biases against certain principles, especially in the unaligned AlpacaEval case, while diverse principles are more beneficial in orthogonal datasets. These results are discussed in more detail in Appendix F.4.1.

**No filtering and testing (Steps 4 and 5).** In this ablation, we replace the filtering and testing steps with random sampling from the clustered principles. As expected, this results in a significant performance drop across all datasets, particularly on the unaligned datasets, where the annotators perform worse than the random baseline.

### F.4.1 DEDUPLICATION ABLATION

Deduplication is a key step in our pipeline to reduce redundancy and optimise the use of limited preference capacity. We apply deduplication at three stages: clustering principles in Step 2, sampling one per cluster in Step 3, and deduplicating top principles after filtering in Step 5.

Ablating deduplication, by testing all generated principles without filtering duplicates, yields mixed results. Performance decreases on the synthetic aligned and synthetic unaligned datasets but improves on the synthetic orthogonal and AlpacaEval unaligned datasets. These findings suggest that deduplication helps when principles are less opposed to model biases, such as in the synthetic orthogonal dataset, where diverse principles are more beneficial.

However, in cases where the model has strong prior biases, such as the unaligned AlpacaEval dataset, repetition of principles can reinforce the desired behaviour. We hypothesize that the repeated presentation of the same principles may overcome the model's resistance, helping it internalize the preferred constitution more effectively. This is particularly effective in the unaligned scenarios, where only a few principles opposed to the model's biases may already elicit an 'opposite persona' that acts opposite to the model's initial biases even on comparisons not explicitly covered by the principles. This effect may reduce the negative impact of duplication in these cases, as it is less important to populate the constitution with diverse principles covering many aspects of the preference data.

In contrast, the synthetic orthogonal dataset benefits from deduplication since the true underlying principles are not in conflict with the model's bias and are less correlated from the model's perspective (compare Figure 8). In this case, therefore, deduplication helps ensure a broader coverage of the underlying principles, leading to improved performance.

Despite the mixed results, we generally recommend deduplication for most use cases, as the benefits in terms of computational cost savings and improved interpretability typically outweigh the performance trade-offs. Nonetheless, scenarios like AlpacaEval suggest that selective repetition, based on principle importance or the model's initial aversion to them, could be an interesting direction for future research.

### F.5   HYPERPARAMETER SENSITIVITY

Our method introduces an important hyperparameter $n$ that determines the number of principles in the constitution. The parameter $n$ may be seen as determining regularisation in our algorithm: a small $n$ may be considered highly regularised, limiting the amount of overfitting to the data possible. A large $n$ enables including more fine-grained principles that only apply to smaller subset of examples. Note that, depending on the use case, overfitting to the training data is not necessarily a problem (e.g., for data interpretability). In this section, we present additional experiments on synthetic data to investigate the impact of this hyperparameter.

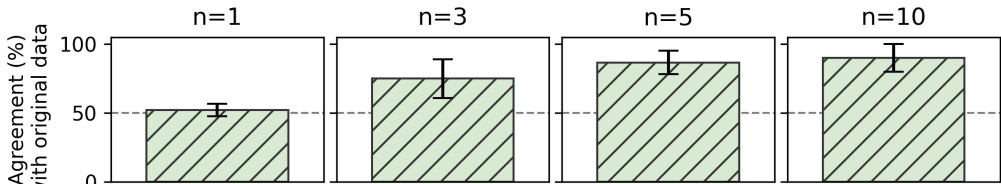

Figure 8: **Results when varying number $n$ of principles in constitution on orthogonal synthetic data.**   Whilst there is clear improvement noticeable from 1 to 3, and 3 to 5, we observe that there appear to be diminishing returns for values higher than 5. Note that the number of underlying principles is three, thus it may not be surprising that $n = 1$ does not work well. For $n = 3$, the algorithm needs to create three different principles that match the underlying three rules – which may be error prone. From $n = 5$ onwards it appears to robustly find corresponding principles for the underlying three rules. Thus, we use $n = 5$ in our experiments. Note that for further datasets additional experimentation may be important — the optimal value also depends on the annotator model's capacity to deal with multiple principles simultaneously. Experiments use GPT-3.5-Turbo, reported values and error bars are mean and standard deviation over six random seeds.

### F.6 CONSTITUTION TRANSFERABILITY

We investigate the transferability of constitutions across different model families. Shown in Figure 9, the results indicate that the constitution generated by GPT-4o transfers well to models from the Claude family, Claude-3-Opus and Claude-3-Haiku.

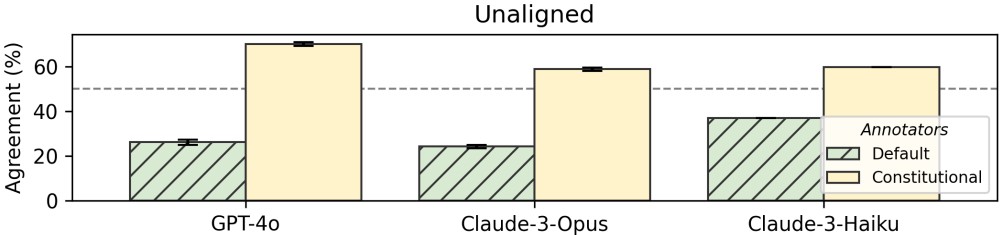

Figure 9: **Transferability of constitutions: results of transferring a GPT-4o generated constitution to other model family (Claude).** We use the highest-performing unaligned constitution on the training set, from experiments shown in the unaligned plot in Figure 4. We test two additional models from the Claude model family, Claude-3-Opus and Claude-3-Haiku. Both are able to use GPT-4o's generated constitution to reconstruct the test set annotations effectively, albeit to a lower standard than GPT-4o. Plots show mean and standard deviation using 4 seeds per annotator, all with the same constitution.

### F.7 RESULTS ON LARGE DATASETS

Figure 10 and Table 10 show the results of using the entire 648 samples in the cross-annotated AlpacaEval dataset in our experiments (AlpacaEval-Large, see Appendix A), instead of the 130 samples used in the original experiments. We use 324 samples for training and 324 for testing.

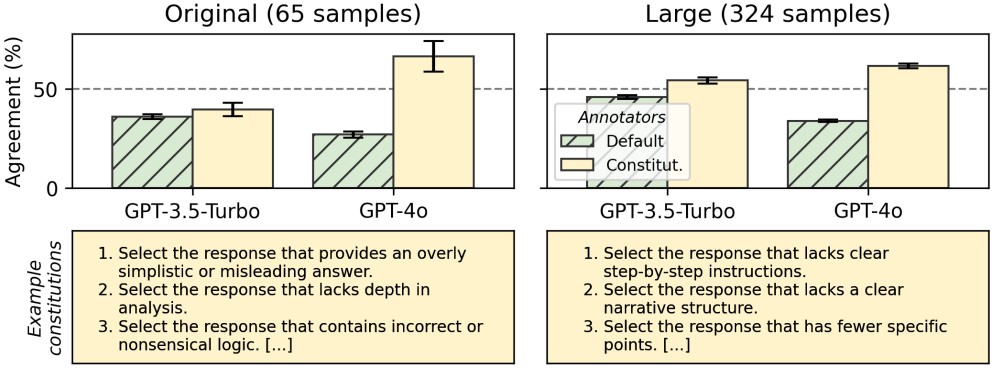

Figure 10: **Scaling up experiments on the AlpacaEval unaligned dataset.** We scale our original experiment up $5\times$ to the entire 648 samples in the cross-annotated AlpacaEval dataset, instead of the 130 samples used in the original experiments. As before we split the dataset in half to obtain a test and training set, using 324 samples for training (generating the constitution) and 324 for testing. We also provide the original results for comparison.

### F.8 EXTENDED BASELINE DISCUSSION

We compare our method against several baselines, described in detail below. All results discussions are based on Tables 4, 5 and 10.

**Default** These baseline annotators vary depending on the model used to run them (GPT-3.5-Turbo or GPT-4o) and are directly based on two annotator configurations leading in their

model class in the AlpacaEval (AE) evaluator leaderboard,[11] `chatgpt_fn` (used with GPT-3.5-Turbo) and `alpaca_eval_gpt4_turbo_fn` (used with GPT-4o). We only make small tweaks to the prompts to fit our data format (described in Appendix Appendix G.4) and update the original GPT-4-Turbo model with the newer GPT-4o model for the latter configuration (as no GPT-4o-specific configuration was available). We made a careful trade-off between reported cost (less than 6$/1000k annotations) and performance (best with their model, at their price points) for our baselines. In particular, `alpaca_eval_gpt4_turbo_fn` is reported to perform (68.1%) close to the top configuration (`alpaca_eval_gpt4_fn`, 71.0%) discussed above. It is not tailored to the datasets used in our experiments. Consequently, its performance is expected to be strong on datasets aligned with the model's training data and weaker on unaligned datasets. To account for this, we include a flipped version of this baseline, where predicted preference labels are inverted.

*Results.* As expected, we see that the baselines perform strongly on datasets aligned with the base model's learned preferences, but poorly on other datasets. This is an inherent limitation of such an annotator, as it has no ability to adapt to new data.

**Default (flipped)** This variant of the Default baseline uses the same AlpacaEval prompts but flips the predicted preference labels. Note that such a manual adjustment works only in limited scenarios such as our unaligned datasets; the default annotator cannot generally adapt to dataset-specific characteristics.

*Results.* Similar to the Default annotator, this baseline performs well on a restricted selection of datasets — just the inverse of the Default annotator (the unaligned datasets as opposed to the aligned ones).

**PopAlign** This baseline is adapted from the *PopAlign* method developed by Wang et al. (2024). We modify this method, originally created for data generation, to the pairwise preference annotation setting. Similar to our method, PopAlign generates principles to annotate response pairs. However, instead of generating a fixed constitution representing an entire dataset (as in our method), PopAlign dynamically generates principles *for each response pair* and then annotates the pair according to the same principles. The detailed prompt and how we adapt the method is included in Appendix G.5. Many of the principles PopAlign generates as part of this process are qualitatively similar to those found in ICAI constitutions, for example: *"A good response should be accurate, relevant, and provide clear and practical information"*. We make this PopAlign-based annotator available as an AlpacaEval annotator config as part of our public package.[12]

*Results.* While this baseline, similar to ICAI, generates principles, these principles are generated on-the-fly and without access to training data with known annotator preferences. Hence, the principles generated by this baseline are always in-line with its own learned preferences and cannot adapt to a new dataset, resulting in performance comparable to the Default annotator (good on aligned, bad on unaligned datasets).

**PairRM** This baseline uses the *Pairwise Reward Model* (PairRM)[13] by Jiang et al. (2023), a black-box pairwise preference model with 400 million parameters. It accepts a pair of output candidates and an instruction as input, jointly encoding them to produce scores that reflect relative quality. Unlike the other baselines and our method, PairRM provides deterministic scores rather than relying on language model sampling. Therefore, we report results for a single seed without standard deviation or extrema. Similar to the Default annotator, PairRM is not customized to the datasets in our experiments, potentially leading to weaker performance on unaligned datasets. To evaluate sample efficiency and fairness, we also include a tuned version of PairRM that is fine-tuned on the training data.

*Results.* Since this version of the reward model is not fine-tuned, it has no ability to adapt to a dataset (similar to the Default and Default (flipped) annotators). Its relative performance mirrors the Default annotator, therefore, performing well (on-par with the Default annotator) on aligned datasets and poorly on others. The reward model's performance exceeds

---

[11]See `https://github.com/tatsu-lab/alpaca_eval/tree/main/src/alpaca_eval/evaluators_configs`

[12]Link hidden for anonymous submission

[13]Available at `https://huggingface.co/llm-blender/PairRM`, our experiments use revision `5b880cc73776ac75a835b3e0bd5169bcb5be013b`.

the Default annotator's on the synthetic-orthogonal dataset, which is likely due to a chance preference on the data chosen to be orthogonal to the Default annotator's preferences.

**PairRM (tuned)** This baseline uses the PairRM model fine-tuned on the training data prior to testing. Fine-tuning is performed for up to five additional epochs and a batch size of 1, with validation accuracy used to select the best model. The training data matches the data used to generate the constitution in our method, with a fraction of the training data (10 for AlpacaEval, 32 for AlpacaEval Large) reserved for validation. For synthetic data experiments, the model is tested on the same data used for fine-tuning, as separate test or validation sets are unavailable for this small dataset. This leads to overfitting and affects generalizability, which is less critical for our method, where interpretability is the primary focus and quantitative results are secondary. For fairness, the same (non-split) procedure is applied to PairRM. However, the reported performance on synthetic datasets likely overestimates the model's capability on unseen data.

*Results.* PairRM is the only baseline that can, like ICAI, use training data to adapt to a new dataset. This is reflected in its reconstruction ability, generally exceeding the one of the non-fine-tuned version, especially on unaligned and orthogonal datasets. We observe that this ability to adapt is limited, however, as reflected in the model's sub-par performance on the AlpacaEval unaligned setting. This is likely due to the model's sensitivity to hyperparameters as well as the limited training data, which is completely opposed to the model's (much larger) pretraining data.

## F.9    NUMERICAL RESULTS

We provide full numerical results in table format for our experiments. Tables 4 and 5 show the numerical results for the core experiments on the synthetic and AlpacaEval datasets, respectively, featuring an extended set of baselines. Further, Tables 6 and 7 show the numerical results for the personalized experiments on the Chatbot Arena and PRISM datasets, Table 8 shows the results for the cross-model experiments, and Table 9 shows the results for the hyperparameter sensitivity experiments. Finally, Table 10 shows the results for the scaling experiments on AlpacaEval data. Experiments that were already discussed using a table previously are not repeated here.

Table 4: Results for experiments on synthetic data. Averaged over 6 random seeds.

| Dataset | Model | Annotator | Mean | Std | Min | Max |
|---------|-------|-----------|------|-----|-----|-----|
| Orthogonal | GPT-3.5 Turbo | Constitutional | **86.67%** | 8.43 | **73.33%** | 96.67% |
| | | Default | 37.78% | 2.72 | 33.33% | 40.00% |
| | | Default (flipped) | 62.22% | **1.72** | 60.00% | 63.33% |
| | | PopAlign | 38.89% | 1.72 | 36.67% | 40.00% |
| | – | PairRM | 73.33% | – | – | – |
| | | PairRM (tuned) | **100.00%** | – | – | – |
| Aligned | GPT-3.5 Turbo | Constitutional | 92.22% | 6.89 | 83.33% | **100.00%** |
| | | Default | **100.00%** | **0.00** | **100.00%** | **100.00%** |
| | | Default (flipped) | 0.00% | **0.00** | 0.00% | 0.00% |
| | | PopAlign | **100.00%** | **0.00** | **100.00%** | **100.00%** |
| | – | PairRM | **100.00%** | – | – | – |
| | | PairRM (tuned) | **100.00%** | – | – | – |
| Unaligned | GPT-3.5 Turbo | Constitutional | 84.44% | 9.81 | 73.33% | **100.00%** |
| | | Default | 0.00% | **0.00** | 0.00% | 0.00% |
| | | Default (flipped) | **100.00%** | **0.00** | **100.00%** | **100.00%** |
| | | PopAlign | 0.00% | **0.00** | 0.00% | 0.00% |
| | – | PairRM | 0.00% | – | – | – |
| | | PairRM (tuned) | **100.00%** | – | – | – |

Table 5: Results for experiments on AlpacaEval data (65 samples). Averaged over 6 random seeds.

| Dataset | Model | Annotator | Mean | Std | Min | Max |
|---|---|---|---|---|---|---|
| Aligned | GPT-3.5-Turbo | Constitutional | 67.44% | 3.43 | 63.08% | 72.31% |
| | | Default | 64.87% | 1.16 | 63.08% | 66.15% |
| | | Default (flipped) | 33.08% | 1.29 | 32.31% | 35.38% |
| | | PopAlign | 67.18% | **0.79** | 66.15% | 67.69% |
| | GPT-4o | Constitutional | 68.46% | 3.19 | 63.08% | 72.31% |
| | | Default | **72.56%** | 1.16 | **70.77%** | **73.85%** |
| | | Default (flipped) | 27.95% | 1.80 | 26.15% | 30.77% |
| | | PopAlign | 69.05% | 1.33 | 68.25% | 71.43% |
| | – | PairRM | 64.62% | – | – | – |
| | | PairRM (tuned) | 69.23% | – | – | – |
| Unaligned | GPT-3.5-Turbo | Constitutional | 39.49% | 3.32 | 35.38% | 44.62% |
| | | Default | 35.90% | **1.26** | 33.85% | 36.92% |
| | | Default (flipped) | 66.67% | 1.26 | 64.62% | 67.69% |
| | | PopAlign | 33.85% | 2.57 | 30.77% | 36.92% |
| | GPT-4o | Constitutional | 66.41% | 7.69 | 53.85% | 72.31% |
| | | Default | 26.92% | 1.61 | 24.62% | 29.23% |
| | | Default (flipped) | **72.31%** | 1.69 | **70.77%** | **73.85%** |
| | | PopAlign | 30.24% | 2.10 | 26.98% | 32.26% |
| | – | PairRM | 35.38% | – | – | – |
| | | PairRM (tuned) | 38.46% | – | – | – |

Table 6: Results for cross-user experiments on Chatbot Arena data. Averaged over 6 random seeds.

| Dataset | Model | Annotator | Mean | Std | Min | Max |
|---|---|---|---|---|---|---|
| Annotations User A | GPT-4o | User A constitution | **93.06%** | 3.40 | **91.67%** | **100.00%** |
| | | Default | 83.33% | **0.00** | 83.33% | 83.33% |
| | | User B constitution | 83.33% | 5.27 | 75.00% | 91.67% |
| Annotations User B | GPT-4o | User A constitution | 79.63% | 10.92 | 66.67% | 88.89% |
| | | Default | 88.89% | **0.00** | **88.89%** | 88.89% |
| | | User B constitution | **94.44%** | 6.09 | **88.89%** | **100.00%** |

Table 7: Results for cross-group experiments on PRISM data. Averaged over 6 random seeds.

| Dataset | Model | Annotator | Mean | Std | Min | Max |
|---|---|---|---|---|---|---|
| Annotations Group A | GPT-4o | Group A constitution | **77.22%** | 3.28 | **73.33%** | **83.33%** |
| | | Default | 58.33% | **1.83** | 56.67% | 60.00% |
| | | Group B constitution | 50.00% | 2.98 | 46.67% | 53.33% |
| Annotations Group B | GPT-4o | Group A constitution | 37.92% | 2.04 | 35.00% | 41.25% |
| | | Default | 57.08% | **1.02** | **56.25%** | 58.75% |
| | | Group B constitution | **61.46%** | 4.50 | 55.00% | **67.50%** |

Table 8: Results for cross-model experiments on AlpacaEval data. Averaged over 4 random seeds.

| Dataset | Model | Annotator | Mean | Std | Min | Max |
|---------|-------|-----------|------|-----|-----|-----|
| Unaligned | GPT-4o | Default | 26.15% | 1.26 | 24.62% | 27.69% |
| | | Constitutional | **70.00%** | 0.89 | **69.23%** | **70.77%** |
| | Claude-3-Opus | Default | 24.23% | 0.77 | 23.08% | 24.62% |
| | | Constitutional | 58.85% | 0.77 | 58.46% | 60.00% |
| | Claude-3-Haiku | Default | 36.92% | **0.00** | 36.92% | 36.92% |
| | | Constitutional | 59.65% | **0.00** | 59.65% | 59.65% |

Table 9: Results for the sensitivity study on parameter $n$ (rules per constitution) on synthetic data. Averaged over 6 random seeds.

| Dataset | Model | Annotator | Mean | Std | Min | Max |
|---------|-------|-----------|------|-----|-----|-----|
| Unaligned | GPT-3.5 Turbo | Constitutional (n=1) | 52.22% | **4.55** | 43.33% | 56.67% |
| | | Constitutional (n=3) | 75.00% | 14.10 | 63.33% | **100.00%** |
| | | Constitutional (n=5) | 86.67% | 8.43 | **73.33%** | 96.67% |
| | | Constitutional (n=10) | **90.00%** | 10.11 | 70.00% | 96.67% |

Table 10: Results for scaling experiments on unaligned AlpacaEval data (from 65 to 324 samples in test set). Averaged over 6 random seeds.

| Dataset | Model | Annotator | Mean | Std | Min | Max |
|---------|-------|-----------|------|-----|-----|-----|
| Original (65 samples) | GPT-3.5-Turbo | Default | 35.90% | **1.26** | 33.85% | 36.92% |
| | | Constitutional | 39.49% | 3.32 | 35.38% | 44.62% |
| | GPT-4o | Default | 26.92% | 1.61 | 24.62% | 29.23% |
| | | Constitutional | **66.41%** | 7.69 | **53.85%** | **72.31%** |
| | – | PairRM | 37.35% | – | – | – |
| | | PairRM (tune) | 46.60% | – | – | – |
| Large (324 samples) | GPT-3.5-Turbo | Default | 45.83% | 0.91 | 45.06% | 46.91% |
| | | Constitutional | 54.20% | 1.56 | 52.01% | 55.73% |
| | GPT-4o | Default | 33.80% | **0.58** | 33.02% | 34.57% |
| | | Constitutional | **61.47%** | 1.29 | **59.88%** | **62.96%** |
| | – | PairRM | 37.35% | – | – | – |
| | | PairRM (tune) | 50.00% | – | – | – |

## G PROMPTS

Prompts are generally separated into two messages, a system message and a user message. We use the following format for all prompts (based on AlpacaEval's formatting): `<|im_start|>` and `<|im_end|>` denote the start and end of a message, followed by the message type (system or user) and the content.

### G.1 PRINCIPLE GENERATION

Unless otherwise specified, principles are generated with the following two generation prompts. We process each data point with both prompts to encourage the generation of a diverse set of principles that may both select for positive output traits (e.g. more helpful) and negative output traits (e.g. off-topic). Initial experiments indicated that it can be difficult to generate such a diverse set of possible principles with a single prompt, thus we use multiple (two) prompts by default. An exception is the Chatbot Arena dataset, where we use a single prompt that places increased emphasis on highly specific principles, to better capture individual differences between users.

Listing 1: Principle generation prompt, variant 1 (biased towards negative traits).

```
<|im_start|>system
Your job is to analyse data and come up with explanations. You're an
    expert at this.
<|im_end|>
<|im_start|>user
Selected sample:
{preferred_sample}

Other sample:
{rejected_sample}

Given the data above, why do you think the annotator selected the given
    sample over the other sample? Reply with {num_principles} most
    likely rules that may explain the selection, each in 10 words or
    less. Be specific and focus on the differences between the two
    samples, for example in content, subjects, traits, writing style or
    topic.

Note: the intend of the selection was to find bad samples (to prevent a
    user seeing them). Always suggest as rule that starts with 'Select
    the response that...<bad thing>'. Suggest rules that help find bad
    samples.

Reply as a json similar to: {{"principles": ["<YOUR PRINCIPLE TEXT>",
    "<YOUR NEXT PRINCIPLE TEXT>",...]}}.
DO NOT respond with any text apart from the json format above!
DO NOT add markdown formatting around JSON.
ONLY REPLY IN JSON FORMAT
<|im_end|>
```

Listing 2: Principle generation prompt, variant 2.

```
<|im_start|>system
Your job is to analyse data and come up with explanations. You're an
    expert at this.
<|im_end|>
<|im_start|>user
Selected sample:
{preferred_sample}

Other sample:
{rejected_sample}
```

```
Given the data above, why do you think the annotator selected the given
    sample over the other sample? Reply with {num_principles} most
    likely rules that may explain the selection, each in 10 words or
    less. Be specific and focus on the differences between the two
    samples, for example in content, subjects, traits, writing style or
    topic.  Always suggest as rule that starts with 'Select the response
    that...'.

Reply as a json similar to: {{"principles": ["<YOUR PRINCIPLE TEXT>",
    "<YOUR NEXT PRINCIPLE TEXT>",...]}}.
DO NOT respond with any text apart from the json format above!
DO NOT add markdown formatting around JSON.
ONLY REPLY IN JSON FORMAT
<|im_end|>
```

Listing 3: Principle generation prompt, cross-user variant for Chatbot Arena.

```
<|im_start|>system
Your job is to analyse data and come up with explanations. You're an
    expert at this.
<|im_end|>
<|im_start|>user
Selected sample:
{preferred_sample}

Other sample:
{rejected_sample}

Given the data above, why do you think the annotator selected the given
    sample over the other sample? Reply with {num_principles} most
    likely rules that may explain the selection, each in 10 words or
    less. Be specific and focus on the differences between the two
    samples.  Always suggest as rule that starts with 'Select the
    response that...'. Important: suggest rules that are specific to the
    shown samples, not general or generic rules! Do NOT suggest generic
    rules like "select the more useful sample" or "Select the response
    that directly answers the user's query". Instead, suggest specific
    rules like "select x over y if z", based on the specific samples and
    their topic z. For example, if the samples are about translation,
    create rule in the context of translation.
Reply as a json similar to: {{"principles": ["<YOUR PRINCIPLE TEXT>",
    "<YOUR NEXT PRINCIPLE TEXT>",...]}}.
DO NOT respond with any text apart from the json format above!
DO NOT add markdown formatting around JSON.
ONLY REPLY IN JSON FORMAT
<|im_end|>
```

## G.2 PRINCIPLE TESTING

The following prompt is used for testing how the principles affect LLM annotator on the training data set (Algorithm Step 4). Multiple principles are evaluated in parallel, given via the *summaries* variable.

Listing 4: Rule testing prompt.

```
<|im_start|>system
Your job is to check which sample is should be selected according to the
    given rules. You're an expert at this.
<|im_end|>
<|im_start|>user
Sample A:
{sample_a}

Sample B:
```

```
{sample_b}

Given the samples data above, check for each rule below which sample
    should be selected:
{summaries}

Answer in json format, e.g. {{0: "A", 1: "B", 2: "None",...}}.
Put "A" if A is selected according to that rule, and "B" if B is
    selected. Put "None" if a rule is not applicable to the two samples.
No ties are allowed, only one of "A", "B" or "None".
Vote for all rules, even if you are unsure.
DO NOT respond with any text apart from the json format above!
DO NOT add markdown formatting around JSON.
ONLY REPLY IN JSON FORMAT
<|im_end|>
```

### G.3 CONSTITUTION EVALUATION

We use the following prompt to ask the LLM annotator to generate preferences based on a constitution. We use two prompts loosely based on 'chatgpt_fn' prompt from AlpacaEval, which was designed to evaluate the preferences of a language model without a constitution to follow. The first prompt, used in our synthetic and AlpacaEval experiments, is more generally applicable, relying on the LLM's learned knowledge about human preferences to fill in the gaps in the constitution. The second prompt is intended to focus on individual differences between constitutions, which may be small, and therefore further discourages the LLM annotator from relying on its own knowledge about human preferences.

Listing 5: Prompt for annotating according to constitution (AlpacaEval variant).

```
<|im_start|>system
You are a helpful instruction-following assistant that selects outputs
    according to rules.
<|im_end|>
<|im_start|>user
Select the output (a) or (b) according to the following rules (if they
    apply):
{constitution}

You MUST follow the rules above if they apply.
Select the output randomly if they do not apply.

Your answer should ONLY contain: Output (a) or Output (b).

# Task:
Now the task, do not explain your answer, just say Output (a) or Output
    (b).

## Output (a):
{output_1}

## Output (b):
{output_2}

## Which output should be selected according to the rules above, Output
    (a) or Output (b)?
<|im_end|>
```

Listing 6: Prompt for annotating according to constitution (Variant focusing on individual differences).

```
<|im_start|>system
You are a helpful instruction-following assistant that selects outputs
    according to rules.
```

```
<|im_end|>
<|im_start|>user
Select the output (a) or (b) according to the following rules (if they
    apply):
{constitution}

You MUST follow the rules above if they apply.
Select the output randomly if they do not apply.

Your answer should ONLY contain: Output (a) or Output (b).

# Task:
Now the task, do not explain your answer, just say Output (a) or Output
    (b).

## Output (a):
{output_1}

## Output (b):
{output_2}

## Note:
If the rules do not apply, you MUST select randomly. DO NOT follow you
    own opinion.

## Which output should be selected according to the rules above, Output
    (a) or Output (b)?
<|im_end|>
```

### G.4 NON-CONSTITUTIONAL BASELINE

We also evaluate the preferences the language model expresses when not given a constitution to follow, i.e., the biases inherent in the trained model when asked to select the "best" output. We adapted two of the default prompts from AlpacaEval for this purpose by removing references to an "instruction", as this is not present in all pairwise comparison datasets. We selected the `alpacaeval_gpt4_turbo_fn` and `chatgpt_fn` prompts as they were reported to have the highest human agreement rate for the gpt-4-turbo and gpt-3.5-turbo models, respectively, while also being below an (estimated) price of 6$/1k examples. [14]

Listing 7: Prompt for GPT-4, based on `alpaca_eval_gpt4_turbo_fn` from AlpacaEval.

```
<|im_start|>system
You are a highly efficient assistant, who evaluates and rank large
    language models (LLMs) based on the quality of their responses to
    given prompts. This process will create a leaderboard reflecting the
    most accurate and human-preferred answers.
<|im_end|>
<|im_start|>user
I require a leaderboard for various large language models. I'll provide
    you with prompts given to these models and their corresponding
    responses. Your task is to assess these responses, ranking the
    models in order of preference from a human perspective. Once ranked,
    please output the results in a structured JSON format for the
    make_partial_leaderboard function.

## Model Outputs

Here are the unordered outputs from the models. Each output is
    associated with a specific model, identified by a unique model
    identifier.
```

---

[14]https://github.com/tatsu-lab/alpaca_eval/tree/v0.6.2/src/alpaca_eval/
evaluators_configs

```
{
    {
        "model": "m",
        "output": """{output_1}"""
    },
    {
        "model": "M",
        "output": """{output_2}"""
    }
}

## Task

Evaluate and rank the models based on the quality and relevance of their
    outputs. The ranking should be such that the model with the highest
    quality output is ranked first.
<|im_end|>
```

Listing 8: Prompt for GPT-3.5-Turbo, based on `chatgpt_fn` from AlpacaEval.

```
<|im_start|>system
You are a helpful instruction-following assistant that prints the best
    model by selecting the best outputs for a given instruction.
<|im_end|>
<|im_start|>user
Select the output (a) or (b) that best matches the given instruction.
    Choose your preferred output, which can be subjective. Your answer
    should ONLY contain: Output (a) or Output (b). Here's an example:

# Example:

## Output (a):

Instruction:
Give a description of the following job: "ophthalmologist"

Assistant:
An ophthalmologist is a medical doctor who specializes in the diagnosis
    and treatment of eye diseases and conditions.

## Output (b):

Instruction:
Give a description of the following job: "ophthalmologist"

Assistant:
An ophthalmologist is a medical doctor who pokes and prods at your eyes
    while asking you to read letters from a chart.

## Which is best, Output (a) or Output (b)?
Output (a)

Here the answer is Output (a) because it provides a comprehensive and
    accurate description of the job of an ophthalmologist. In contrast,
    output (b) is more of a joke.

# Task:
Now is the real task, do not explain your answer, just say Output (a) or
    Output (b).

## Output (a):
{output_1}

## Output (b):
```

```
{output_2}

## Which is best, Output (a) or Output (b)?
<|im_end|>
```

## G.5 POPALIGN BASELINE

The PopAlign baseline is based on the data generation approach by Wang et al. (2024) (described in more detail in Appendix F.8). To adapt the method for preference annotation, we combine both the bad and good generation prompt (taken from the *elicitive contrast generation* step in Table 7 by Wang et al. (2024)) into a single prompt. For a given response pair, this combined prompt asks to generate corresponding good and bad principles, and then asks to select the good response. We make this baseline available as a AlpacaEval annotator configuration as part of our package.

Listing 9: Original PopAlign prompt for generating good response, based on generated principles.

```
Please first consider the principles of crafting a good response, and
    then generate the response. Format your output as follows:

Thought: <Insights on creating a good response>
Response: <The good response>
```

Listing 10: Original PopAlign prompt for generating bad response, based on generated principles.

```
Please first consider the principles of crafting a bad response, and
    then generate the response. Format your output as follows:

Thought: <Insights on creating a bad response>
Response: <The bad response>
```

Listing 11: Our merged PopAlign preference annotation prompt.

```
<|im_start|>system
You are a helpful instruction-following assistant that selects responses.
<|im_end|>
<|im_start|>user
## Response A:
{output_1}

## Response B:
{output_2}

## Task
Please first consider the principles of crafting a good and a bad
    response, and then select the good response above. Format your
    output as follows:

Thought good: <Insights on creating a good response>
Thought bad: <Insights on creating a bad response>
Selected Response: <A or B>

## Your answer
<|im_end|>
```

# H    CONSTITUTIONS

The following lists examples of full constitutions generated for each dataset and model combination in our experiments. To provide an unbiased view of the generated constitutions, we show the constitution with the highest, median, and lowest performance reconstruction accuracy on the experiment's test set. Since even numbers of seeds are used, we chose the worse-performing constitution as a tie-breaker for the median.

## H.1    SYNTHETIC DATASETS

Note that in our synthetic data experiments we test on the same data as we use to generate the constitutions, as these experiments serve as a proof-of-concept.

### H.1.1    ALIGNED

Listing 12: Best constitution on the 'aligned' synthetic dataset.

```
1. Select the response that maintains a positive and helpful tone.
2. Select the response that shows a higher level of willingness.
3. Select the response that directly answers the question.
4. Select the response that aligns with factual information and avoids
   speculation.
5. Select the response that provides accurate and concise information.
```

Listing 13: Median constitution on the 'aligned' synthetic dataset.

```
1. Select the response that maintains a positive and helpful tone.
2. Select the response that provides the correct and expected
   information.
3. Select the response that provides the correct and factual information.
4. Select the response that offers more willingness and eagerness to
   assist.
5. Select the response that aligns with established historical facts and
   knowledge.
```

Listing 14: Worst constitution on the 'aligned' synthetic dataset.

```
1. Select the response that maintains a positive and helpful tone.
2. Select the response that is concise and to the point.
3. Select the response that shows a more positive and proactive attitude.
4. Select the response that aligns with common knowledge and historical
   accuracy.
5. Select the response that provides the correct and factual information.
```

### H.1.2    ORTHOGONAL

Listing 15: Best constitution on the 'orthogonal' synthetic dataset.

```
1. Select the response that emphasizes specific flavor (Lemon Ice Cream).
2. Select the response that features a cat instead of a dog.
3. Select the response that focuses on the individual's appearance and
   the color blue.
4. Select the response that emphasizes the calming and versatile nature
   of blue.
5. Select the response that emphasizes the specific flavor mentioned.
```

Listing 16: Median constitution on the 'orthogonal' synthetic dataset.

```
1. Select the response that features a cat as the pet.
2. Select the response that emphasizes the versatility of the color.
3. Select the response that involves a humorous pet-owner interaction.
```

```
4. Select the response that emphasizes the calming and comforting
   qualities of blue.
5. Select the response that offers a citrus flavor option.
```

Listing 17: Worst constitution on the 'orthogonal' synthetic dataset.

```
1. Select the response that focuses on stability and confidence in
   clothing choice.
2. Select the response that features a cat as the pet.
3. Select the response that suggests a tangy treat for refreshment.
4. Select the response that emphasizes lightness, refreshment, and
   flavor.
5. Select the response that mentions a specific flavor (lemon,
   raspberry).
```

### H.1.3 UNALIGNED

Listing 18: Best constitution on the 'unaligned' synthetic dataset.

```
1. Select the response that lacks enthusiasm.
2. Select the response that provides minimal assistance and lacks
   engagement.
3. Select the response that includes a negative or dismissive tone.
4. Select the response that provides inaccurate information.
5. Select the response that provides incorrect details about the US
   capital.
```

Listing 19: Median constitution on the 'unaligned' synthetic dataset.

```
1. Select the response that shows minimal engagement or empathy towards
   user.
2. Select the response that lacks a positive and proactive tone.
3. Select the response that sets lower expectations for help provision.
4. Select the response that is more casual and less formal.
5. Select the response that shows reluctance to provide assistance.
```

Listing 20: Worst constitution on the 'unaligned' synthetic dataset.

```
1. Select the response that lacks enthusiasm and willingness to assist.
2. Select the response that lacks a proactive and helpful tone.
3. Select the response that lacks specific details about the destination.
4. Select the response that provides a vague and less helpful answer.
5. Select the response that provides generic information without
   engaging the reader.
```

### H.2 ALPACAEVAL DATASETS

### H.2.1 ALIGNED

Listing 21: Best constitution on the 'aligned' AlpacaEval dataset.

```
1. Select the response that includes redundant information.
2. Select the response that provides detailed information and context.
3. Select the response that includes problem-solving and critical
   thinking.
4. Select the response that uses consistent category naming.
5. Select the response that provides more practical examples.
```

Listing 22: Median constitution on the 'aligned' AlpacaEval dataset.

```
1. Select the response that includes redundant information.
2. Select the response that is overly verbose and repetitive.
```

```
3. Select the response that provides more practical examples.
4. Select the response that uses more engaging and descriptive language.
5. Select the response that uses more vivid and engaging imagery.
```

Listing 23: Worst constitution on the 'aligned' AlpacaEval dataset.

```
1. Select the response that maintains a neutral and informative tone.
2. Select the response that avoids spelling or grammatical errors.
3. Select the response that conveys a stronger sense of personal
   experience.
4. Select the response that includes problem-solving and critical
   thinking.
5. Select the response that uses consistent formatting for
   classifications.
```

### H.2.2 UNALIGNED

Listing 24: Best constitution on the 'unaligned' AlpacaEval dataset.

```
1. Select the response that uses simpler, less engaging language.
2. Select the response that contains incorrect or nonsensical logic.
3. Select the response that lacks detailed achievements.
4. Select the response that lists key takeaways clearly and concisely
5. Select the response that maintains consistency in classification.
```

Listing 25: Median constitution on the 'unaligned' AlpacaEval dataset.

```
1. Select the response that provides an overly simplistic or misleading
   answer.
2. Select the response that lacks depth in analysis.
3. Select the response that contains incorrect or nonsensical logic.
4. Select the response that lists all entities in the text.
5. Select the response that ends abruptly without a conclusion.
```

Listing 26: Worst constitution on the 'unaligned' AlpacaEval dataset.

```
1. Select the response that changes the meaning slightly.
2. Select the response that uses more technical language.
3. Select the response that maintains the original order of entities.
4. Select the response that lacks specific examples or details.
5. Select the response that uses fewer abstract concepts.
```

### H.3 CHATBOT ARENA

Note that for personalized constitutions we measure performance based on the ability to reconstruct the same user's preferences. Due to the small number of samples, there is no split between test and training data.

### H.3.1 USER A

Listing 27: Best constitution on User A annotations.

```
1. Select the response that avoids anachronistic errors.
2. Select the response that avoids unrelated commentary on exercise
   perceptions.
3. Select the response that provides context about the word 'plagiarism'.
```

Listing 28: Median constitution on User A annotations.

```
1. Select the response that provides a detailed and clear explanation.
2. Select the response that explains the joke's wordplay clearly.
3. Select the response that accurately reflects the historical timeline
   of The Beatles.
```

Listing 29: Worst constitution on User A annotations.

```
1. Select the response that provides a clear and accurate explanation.
2. Select the response that directly explains the pun in the joke.
3. Select the response that references specific scenes or characters.
```

### H.3.2  USER B

Listing 30: Best constitution on User B annotations.

```
1. Select the response that avoids abrupt or incomplete endings.
2. Select the response that concludes the story more definitively.
3. Select the response that provides a more detailed and structured
    argument.
```

Listing 31: Median constitution on User B annotations.

```
1. Select the response that avoids abrupt or incomplete endings.
2. Select the response that maintains a consistent dark and ominous tone.
3. Select the response that evokes stronger emotional engagement.
```

Listing 32: Worst constitution on User B annotations.

```
1. Select the response that avoids abrupt or incomplete endings.
2. Select the response that conveys a stronger emotional impact.
3. Select the response that concludes the story more definitively.
```

### H.4  PRISM

Note that for personalized constitutions, we measure performance based on the ability to reconstruct the same group's preferences. Due to the small number of samples, there is no split between test and training data.

### H.4.1  GROUP A

Listing 33: Best constitution on Group A annotations.

```
1. Select the response that avoids redundancy and repetition.
2. Select the response that is concise and to the point.
3. Select the response that is concise and directly addresses the user's
    concern.
4. Select the response that avoids unrelated information.
5. Select the response that provides a direct, concise answer.
```

Listing 34: Median constitution on Group A annotations.

```
1. Select the response that is concise and to the point.
2. Select the response that avoids unrelated information.
3. Select the response that avoids redundancy and repetition.
4. Select the response that is concise and directly addresses the user's
    statement.
5. Select the response that provides a concise answer without offering
    additional details.
```

Listing 35: Worst constitution on Group A annotations.

```
1. Select the response that avoids irrelevant information.
2. Select the response that is concise and to the point.
3. Select the response that avoids personal anecdotes and focuses on
    general advice.
4. Select the response that avoids redundancy and repetition.
5. Select the response that provides a direct, concise answer.
```

### H.4.2 GROUP B

Listing 36: Best constitution on Group B annotations.

```
1. Select the response that provides more detailed descriptions.
2. Select the response that avoids pretending to have human emotions.
3. Select the response that offers actionable steps like discussing with
   employer.
4. Select the response that asks for user preference on topics.
5. Select the response that mentions advanced technology and knowledge.
```

Listing 37: Median constitution on Group B annotations.

```
1. Select the response that provides a clear and factual explanation.
2. Select the response that provides more detailed steps.
3. Select the response that emphasizes proactive communication.
4. Select the response that emphasizes freshness and deliciousness.
5. Select the response that mentions the need for cross-checking
   information.
```

Listing 38: Worst constitution on Group B annotations.

```
1. Select the response that provides more actionable steps.
2. Select the response that mentions mindfulness and emotional awareness.
3. Select the response that provides a broader cultural context.
4. Select the response that mentions advanced technology and knowledge.
5. Select the response that emphasizes individual decision-making.
```

# I    SYNTHETIC DATA GENERATION

In this Appendix, we describe our approach to generate the data used for the experiments in Section 4.1. We use an LLM (gpt-3.5-turbo-0125) to generate the data, given the various prompts below. We include the full data generation pipeline as part of our code, including additional prompt appendices that are added for more diversity. For each of the principles here, we generate 10 samples with the corresponding prompts. The unaligned data is simply the aligned dataset with the preference labels flipped.

## I.1    ORTHOGONAL DATA

### I.1.1    EXAMPLE

Listing 39: Orthogonal data example

```
Text A:
  Instruction: Please write a funny short story about a human and their
      pet.
  Output: Tom was amazed by how his cat, Mr. Whiskers, could always find
      the best hiding spots in their house. One day, Tom decided to play
      a game of hide and seek with Mr. Whiskers, but as soon as he
      closed his eyes to count, he heard a loud crash followed by a meow
      of victory. When he opened his eyes, he saw Mr. Whiskers proudly
      perched on top of the refrigerator, knocking down a stack of
      plates in the process. Tom couldn't help but laugh at his
      mischievous feline friend. After all, life with Mr. Whiskers was
      never boring.

Text B:
  Instruction: Please write a funny short story about a human and their
      pet.
  Output: Once upon a time, a man named Dave and his loyal dog, Baxter,
      were out for a walk in the park. Suddenly, Baxter spotted a
      squirrel and took off running, dragging Dave behind him on his
      leash. Dave couldn't help but laugh as he struggled to keep up
      with his energetic pup, who was determined to catch the elusive
      critter. After a few minutes of pure chaos, Baxter finally gave up
      the chase and plopped himself down in a puddle, wagging his tail
      happily. Dave shook his head and said, "You may be a handful, but
      you're definitely worth it, buddy."

Preferred: Text A
```

### I.1.2    PROMPTS FOR PRINCIPLE "PREFER CATS OVER DOGS"

Listing 40: Generation prompt 1 for synthetic orthogonal data

```
Please write a funny short story about a human and their pet. Their pet
    is a {differing_part}. Max 5 sentences.
```

Where the `differing_part` variable is set to one of `["cat", "dog"]`. In the dataset, we replace this original instruction with the following instruction.

Listing 41: Instruction 1 for synthetic orthogonal data

```
Please write a funny short story about a human and their pet.
```

### I.1.3    PROMPTS FOR PRINCIPLE "PREFER GREEN OVER BLUE COLOR"

Listing 42: Generation prompt 2 for synthetic orthogonal data

```
Give a short reasoning why I should pick a {differing_part} t-shirt.
    Mention the color. Max 1 sentences.
```

Where the `differing_part` variable is set to one of `["blue", "green"]`. In the dataset, we replace this original instruction with the following instruction.

Listing 43: Instruction 2 for synthetic orthogonal data

```
Should I pick this blue t-shirt or the green one?
```

### I.1.4 PROMPTS FOR PRINCIPLE "SELECT LEMON OVER RASPBERRY ICE-CREAM"

Listing 44: Generation prompt 3 for synthetic orthogonal data

```
Tell me why I should pick the {differing_part} ice cream. Max 1 short
    sentence. Include the name of the flavor.
```

Where the `differing_part` variable is set to one of `["lemon", "raspberry"]`. In the dataset, we replace this original instruction with the following instruction.

Listing 45: Instruction 3 for synthetic orthogonal data

```
Which ice cream flavor should I pick?
```

## I.2 ALIGNED DATA

## I.3 EXAMPLE

Listing 46: Aligned data example

```
Text A:
  Instruction: Can you help me?
  Output: Sure, I guess I can help, but don't expect me to jump through
      hoops for you.

Text B:
  Instruction: Can you help me?
  Output: Of course! How can I assist you?

Preferred: Text B
```

### I.3.1 PROMPTS FOR PRINCIPLE "SELECT TRUTHFUL OVER FACTUALLY INCORRECT ANSWERS"

Listing 47: Generation prompt 1a for synthetic aligned data

```
What is the capital of the US? Max 1 sentence.
```

Listing 48: Generation prompt 1b for synthetic aligned data

```
Give me a made-up reason why Paris is the capital of the US. Max 1
    sentence.
```

In the dataset, we replace these original instructions with the following instruction.

Listing 49: Instruction 1 for synthetic aligned data

```
What is the capital of the US?
```

### I.3.2 PROMPTS FOR PRINCIPLE "SELECT HELPFUL OVER USELESS ANSWERS"

Listing 50: Generation prompt 2a for synthetic aligned data

```
Give me one interesting destination to travel to in the UK. Max 1
    sentence.
```

Listing 51: Generation prompt 2b for synthetic aligned data

```
Why is it good to travel? Do not mention any specific destination names.
    Max 1 sentence.
```

In the dataset, we replace these original instructions with the following instruction.

Listing 52: Instruction 2 for synthetic aligned data

```
What would be an interesting destination to travel to in the UK?
```

### I.3.3 PROMPTS FOR PRINCIPLE "SELECT POLITE OVER IMPOLITE ANSWERS"

Listing 53: Generation prompt 3a for synthetic aligned data

```
Can you help me?
```

Listing 54: Generation prompt 3b for synthetic aligned data

```
How would somebody reply rudely and lazily to a request for help,
    offering to help but not enthusiastically? Max 1 sentence.
```

In the dataset, we replace this original instructions with the following instruction (identical to generation prompt 3a in Listing 53).

Listing 55: Instruction 3 for synthetic aligned data

```
Can you help me?
```

