# OpenReview forum: "Inverse Constitutional AI: Compressing Preferences into Principles"
_ICLR.cc/2025/Conference — ICLR 2025 Poster_

### Official Review · Reviewer_HeqU · 2024-11-03

**Soundness:** 3
**Presentation:** 3
**Contribution:** 4
**Rating:** 8
**Confidence:** 4

**Summary:**

It introduces a novel approach for understanding and interpreting pairwise preference data used in training and evaluating AI models. Traditional methods often use feedback data like pairwise text preferences to align models with human preferences, but they do not explain why one model is preferred over another. This gap in interpretability poses challenges, particularly when biases in human feedback influence model training and evaluation.

To address this, the authors propose the Inverse Constitutional AI (ICAI) problem, which involves extracting a set of natural language principles (a "constitution") from existing feedback data. This set of principles is intended to help a large language model (LLM) reconstruct the original annotations, effectively compressing complex preference data into an interpretable and concise format. The ICAI method could help reveal underlying annotator biases, provide a clearer understanding of model behaviors, and facilitate the creation of customized models aligned with individual or group preferences.

The paper outlines an ICAI algorithm with five main steps: generating candidate principles, clustering similar principles, deduplicating principles, testing principles for their effectiveness in reconstructing feedback, and filtering out less effective principles. The method is tested on synthetic datasets, human-annotated AlpacaEval data, user-specific data from Chatbot Arena, and demographic group data from the PRISM dataset. The experiments show that the generated constitutions can effectively compress and explain preference data, revealing biases and guiding models toward interpretable decision-making.

**Strengths:**

1. One of the standout strengths of the ICAI approach is its ability to convert complex, often opaque preference data into a set of clear, natural language principles. This enhances the interpretability of AI training and evaluation processes, allowing researchers and practitioners to understand the rules and biases underlying model behavior. Such transparency is especially valuable when assessing why certain outputs are favored, which can inform better decision-making and trust in AI systems.

2. The method provides a powerful tool for detecting potential biases embedded in human-annotated feedback. By distilling preferences into principles, ICAI helps identify systematic biases (e.g., preferences for assertiveness over truthfulness) that might not be evident from raw data alone. This can lead to more balanced and fair training processes and better-aligned models.

3. The algorithm's ability to scale feedback data into concise, human-readable principles means it can be adapted for various use cases, including creating personal or group-specific constitutions. This adaptability supports the customization of LLMs to align with individual user preferences or demographic group values, potentially improving user satisfaction and model alignment in diverse contexts.

4. The paper demonstrates that ICAI is applicable to a range of datasets, from synthetic data with known rules to complex, real-world datasets like AlpacaEval, Chatbot Arena, and PRISM. This versatility shows that ICAI can work in controlled experiments as well as in more unpredictable, user-driven scenarios.

**Weaknesses:**

1. One inherent limitation of the ICAI method is that it simplifies complex human annotations into a smaller set of principles, which can result in a lossy representation. This means that the constitution may not capture all nuances of the original data, potentially omitting subtle preferences or context-specific details that influence human judgments. As a result, the reconstructed preferences might not fully align with the complexity of human decision-making.

2. The effectiveness of ICAI heavily depends on how well an LLM can interpret and apply the generated principles. If the LLM misinterprets or inconsistently applies the principles, the reconstructed annotations might diverge from the original data. This dependence introduces variability based on the choice and capability of the LLM used, potentially limiting the generalizability of the approach across different models.

3. The generated principles, while human-readable, may be ambiguous or open to interpretation. This ambiguity can lead to inconsistent applications of the principles, especially when dealing with edge cases or scenarios that the principles do not explicitly address. The method may struggle to create highly precise and unambiguous rules that cover all relevant aspects of the original annotations.

**Questions:**

1. How well do the principles generated by ICAI transfer across different models and datasets? Can the constitutions created for one dataset be adapted effectively for use with other types of preference data?

2. How effective is ICAI at identifying subtle or less obvious biases in preference data? What specific types of biases are more likely to be detected with this approach, and which may be missed?

3. How might ICAI be extended to work with multimodal data (e.g., combining text with images or audio) or more complex preference structures beyond pairwise comparisons?

---

> ### Author Response · Authors · 2024-11-20
>
> Thank you for your positive and detailed feedback! Below, we address each of your points, abbreviating weakness with "W" and question with "Q". References refer to the revised manuscript, where changes are highlighted in blue.
>
> ### W1: One inherent limitation of the ICAI method is that it simplifies complex human annotations into a smaller set of principles, which can result in a lossy representation.
>
> We agree that lossy compression is a limitation of the ICAI method, inherent to the goal of summarizing complex feedback data into a concise and interpretable format. While future work may further refine the method to capture more nuanced preferences, we view this as a trade-off between interpretability and complexity. We believe that the interpretability gained from the concise principles outweighs the loss of some nuanced information. Further, note that on datasets which are mostly aligned with the data used to train the AI annotator, the annotator can fill in some nuances that are lost in the constitution, as long as this judgement does not contradict the constitution. We have added a discussion of to the limitations section (Section 6) to clarify this point.
>
> ### W2: The effectiveness of ICAI heavily depends on how well an LLM can interpret and apply the generated principles.
>
> While true, this reliance can also be a strength. The LLM can leverage prior knowledge to fill in nuances and weigh principles like a human judge. This does come at the cost of interpretability and depends on the LLM's generalization ability. Nonetheless, our experiments showed that ICAI's principles effectively guided the LLM to reconstruct annotations, even on unseen data.
>
> ### W3: The generated principles, while human-readable, may be ambiguous or open to interpretation.
>
> We acknowledge this risk but view it as a double-edged sword. The flexibility of language-based principles enhances expressivity, enabling compact and interpretable preference representations. However, ambiguity can lead to inconsistent applications, particularly in edge cases not explicitly covered by the principles.
>
> ### Q1: How well do the principles generated by ICAI transfer across different models and datasets?
>
> Excellent question!
> Transferability is crucial, as it reflects generalizable principles rather than model-specific artifacts. In Appendix C.5, we show that constitutions generated by GPT-4o transfer effectively to other models (e.g., Claude-3 variants). For datasets, transferability depends on similarity. While constitutions generalize well within the same distribution (e.g., AlpacaEval and PRISM), they may not generalize across distinct distributions, as seen in cross-user and demographic experiments (Sections 4.3 and 4.4).
>
> ### Q2: How effective is ICAI at identifying subtle or less obvious biases in preference data?
>
> We address the use-case of bias detection in the new Section 4.5 (and Appendix C.2) of the updated paper, where we demonstrate how ICAI can be used to uncover and evaluate biases in the Alpaca Eval, Chatbot Arena and PRISM datasets.  While the biases we uncover there are relatively simple, we have added a discussion of how ICAI could be extended to detect more subtle biases in Appendix C.2.
>
> ### Q3: How might ICAI be extended to work with multimodal data or more complex preference structures beyond pairwise comparisons?
>
> We believe that ICAI could straightforwardly be extended to work with multimodal data by adapting the method to generate principles from more complex preference data. An important question to answer would be, however, if a textual representation of the principles is still sufficient in more complex modalities such as audio or video, or if a different representation (possibly in those same modalities) would be more appropriate.
>
> Similarly, more complex preference structures such as ranking or rating data could be addressed by adapting principle generation prompts, as long as the underlying LLM is capable enough to understand and apply these more complex structures -- i.e., interpreting a full ranking may prove challenging due to the long context required. We appreciate that you raise these points, we would be excited for future work to explore both of these questions.
>
> We hope that we could address your concerns and questions effectively. Please let us know if you have any further questions or comments. Thank you for your detailed review!

---

### Official Review · Reviewer_nRv6 · 2024-11-04

**Soundness:** 2
**Presentation:** 3
**Contribution:** 2
**Rating:** 5
**Confidence:** 4

**Summary:**

Paper proposes a framework for interpreting preference datasets used to align large language models (LLMs) with human-like decision-making. ICAI inverts the process of constitutional AI. Rather than using a predefined constitution to guide model behavior, ICAI attempts to derive such principles from preference data. They tested constructed principles by reconstructing preference annotations.

**Strengths:**

1. Developing constitutional principles from feedback data is an important research problem to build an interpretable preference learning framework.

2. This alogrithm is tested on four datasets with synthetic setting, human annotated data, individual user preferences and group preferences.

**Weaknesses:**

1.  Without establishing causality between the principles and annotator rationale, the framework risks over-simplifying or even misrepresenting the underlying preferences. For example, it is possible that the principles reflect incidental biases of the model or dataset rather than genuine human values. This could lead to misleading interpretations and false assumptions about user or demographic intentions.

2. ICAI's approach inherently admits multiple valid constitutions for the same dataset, depending on clustering and sampling choices. This non-uniqueness implies that each run could yield different principles that still achieve similar reconstruction accuracy. This hurts interpretation. Also ICAI seems to be influenced by initial prompt or clustering parameters as well, making it more unstable.

3. The paper primarily focuses on preference reconstruction, yet practical applications, such as bias detection, model debugging, or customization, are only discussed in passing without concrete evidence of their effectiveness. There is no emperical evidence of ICAI's practical application

4. This framework may amplify biases present in the training data by distilling these biases into high-level principles. The paper does not discuss or test for scenarios where harmful biases (such as gender or racial biases) could be encoded into the constitution, which may reinforce harmful stereotypes or skewed preferences.

**Questions:**

See weaknesses

---

> ### Author Response · Authors · 2024-11-20
>
> Thank you for your detailed review! Below, we discuss each point raised. We abbreviate weakness with "W" and question with "Q". All references refer to the revised manuscript, in which we highlight changes in blue.
>
> ### W1 (1): Without establishing causality between the principles and annotator rationale, the framework risks over-simplifying or even misrepresenting the underlying preferences.
>
> We agree that the results need to be interpreted with great care. We discuss the causality limitation and how it affects the usability of our method in the limitations section and hope to clarify this discussion further in this rebuttal.  The alternative to ICAI for most datasets is to simply use them as a black-box --- without having any explanation of what the annotator rationale may have been. We argue that ***some* indications of annotator rationale (even if imperfect) are preferable to *none*.** Please let us know if you think our limitations section does not adequately address this concern.
>
> ### W1 (2): For example, it is possible that the principles reflect incidental biases of the model or dataset rather than genuine human values. This could lead to misleading interpretations and false assumptions about user or demographic intentions.
>
> We agree that misuse of ICAI to create misleading interpretations of annotator values is a concern, as we state as part of our Ethics Statement. Yet, regardless of the annotators, if a dataset happens to have an incidental bias (e.g. stylistic, subconscious or even by chance), then awareness of such a bias is critical --- even if the bias is incidental does not reflect the annotators' values. Awareness can help avoid optimizing towards the bias, either by modifying the constitution or filtering the dataset. We have added a discussion on possible mitigation strategies in the new Appendix C.2, which hopefully supports the benefits of ICAI in detecting and mitigating biases.
>
> ### W2 (1): ICAI's approach inherently admits multiple valid constitutions for the same dataset, depending on clustering and sampling choices. This non-uniqueness implies that each run could yield different principles that still achieve similar reconstruction accuracy. This hurts interpretation.
>
> We agree that non-uniqueness of constitutions and principles makes interpretation more difficult, but this issue is not unique to ICAI but rather generally affects the problem of explaining annotator decisions in natural language. There are numerous ways any given set of principles can be rewritten such that an annotator would come to the same conclusion. Nevertheless, qualitatively there are many similarities in constitutions generated for different seeds and, as discussed above, we would argue that some indications on annotator rational (even if imperfect) are preferable to none.
>
> ### W2 (2): Also ICAI seems to be influenced by initial prompt or clustering parameters as well, making it more unstable.
>
> We agree that ICAI is influenced by its parameters. We provide a small study on hyperparameter sensitivity in appendix C.4. As general advice to mitigate instability, the results of our scaling experiments (shown in Table 4 in Appendix C.6) indicate that running on larger datasets reduces the overall variance of our method. This effect can be seen in terms of a reduction of standard deviation of results as scale goes up.
>
> ### W3: The paper primarily focuses on preference reconstruction, yet practical applications, such as bias detection, model debugging, or customization, are only discussed in passing without concrete evidence of their effectiveness. There is no emperical evidence of ICAI’s practical application.
>
> We agree that the paper could benefit from more concrete evidence of ICAI's practical applications. To fill this gap, **we have added a new set of experiments in Section 4.5 and Appendix C.2, where we use ICAI to detect biases in multiple preference datasets.** We find that ICAI can be used to detect biases in the data. Concretely, we find evidence of verbosity bias, list bias and assertiveness bias in the datasets. We additionally add a discussion of possible bias mitigation strategies in Appendix C.2, which we hope will further support the practical applications of ICAI.

---

> > ### Author Response · Authors · 2024-11-20
> >
> > ### W4: This framework may amplify biases present in the training data by distilling these biases into high-level principles. The paper does not discuss or test for scenarios where harmful biases (such as gender or racial biases) could be encoded into the constitution, which may reinforce harmful stereotypes or skewed preferences.
> >
> > In general, we agree that the risk of amplifying harmful biases should be carefully considered, especially when using human-annotated preference data. However, we believe **our framework can play a vital role in highlighting harmful biases and mitigating their impact.** Most conventional use-cases of preference data hide such biases. For example, both black-box reward models and aggregate evaluation statistics can encode such biases in a way that is hard to detect. ICAI can be applied to the underlying preference data to highlight harmful biases that may transfer to these downstream use-cases. We demonstrate this use of ICAI for bias detection in a new set of experiments discussed in the new Section 4.5 and Appendix C.2. Besides style biases, this analysis finds that Chatbot Arena may have a bias against neutral responses unlike PRISM, which has a bias towards more neutral responses. We also discuss possible mitigation strategies in that section, which can hopefully combat reinforcement of harmful stereotypes.
> >
> > In general, it is much more difficult to hide harmful biases in plain-text constitutions than in black-box reward models or aggregate statistics. Thank you for raising this point, we have added a clarification to our ethics statement to highlight this aspect.

---

> > ### Comment · Reviewer_nRv6 · 2024-11-26
> > **Reply to authors**
> >
> > Thanks for your clarification and response. I will maintain the current scores, as I believe they adequately reflect the quality and contribution of your work.

---

> ### Author Response · Authors · 2024-11-27
>
> Thank you again for your detailed review, and for taking the time to read and consider our response. We would like to ask for your feedback on potential further improvements to the paper. We are currently working on a second revision, which we plan to share within the editing window (27 Nov 11:59pm AoE). Below, we go through your concerns again, summarize how we initially addressed the concerns, and share additional improvements we are currently working on.
>
> - **Constitutions may over-simplify and misrepresent annotator intention (W1):** We adapted our limitations section (6) to further highlight the related tension between interpretability, necessitating a short and possibly over-simplifying constitution, and accuracy.
>   - *__Further planned improvements:__*
>     1. *Expand our ethics statement to especially highlight the raised risk of misrepresentation.*
>     2. *Add corresponding warning to our code output. Both these steps aim to ensure users are fully aware of this limitation and prevent misinterpretation of ICAI results.*
>
> - **Non-uniqueness and variability of constitutions (W2):** We agreed with your concern and recognize it as a fundamental challenge with many interpretability methods like ours. We highlighted that for down-stream applications (like harmful bias detection), finding concerning principles is worthwhile even if other possible explanations exist.
>   - *__Further planned improvements:__*
>     1. *Add in-depth discussion addressing the impact of this limitation as appendix.*
>
> - **Missing practical applications evidence (W3):**
>   We introduced new experiments (Section 4.5, Appendix C.2) to demonstrate ICAI's utility in detecting biases in real-world preference datasets. These experiments show how ICAI can address issues like verbosity and fairness, improving its practical relevance.
>   - *__Further planned improvements:__*
>     1. *We have completed an additional use-case study and will add this to the manuscript, providing further evidence for the usefulness of ICAI to scale annotation data from a small initial dataset, applied to helpful/harmless annotations.*
>
> - **Risk of bias amplification (W4):** We recognized the risk that our method may amplify biases, but we also highlighted ICAI's potential role in detecting, rather than amplifying, harmful biases and provided examples of possible mitigation strategies. These are reflected in the new experiments (Section 4.5 and Appendix C.2) and further detailed in the ethics section.
>   - *__Further planned improvements:__*
>     1. *Expand discussion of ethics section further to more explicitly highlight the risks of our method in terms of bias amplification and provide actionable steps users can take to mitigate this risk (e.g., through manual inspection of constitutions).*
>     2. *Add a corresponding warning to our code output (incl. actionable mitigation steps). Both these steps will ensure users are fully aware of this risk and can mitigate it as far as possible.*
>
> Like any methodology, our approach is not without remaining challenges, but your feedback has helped us refine and clarify these. We believe the changes made in this revision, as well as the planned additional updates outlined above, significantly strengthen the manuscript and address the main issues raised in your review.
>
> We plan to share a second revision of the manuscript soon but, given the limited time remaining in the editing window, wanted to give you the opportunity to share your thoughts on the planned changes and on any other improvements we could make to further strengthen the paper and better address your concerns.
>
> Thank you again for your constructive feedback - it has helped us improve the paper considerably!

---

> > ### Author Response · Authors · 2024-11-28
> >
> > We have completed the second revision, implementing all previously outlined changes. Modifications are highlighted in green and include:
> >
> > - **W1 (Misrepresentation of annotator intention):** We have added a dedicated discussion of this issue to the ethics statement and incorporated warnings to the code output to ensure users are aware of this limitation.
> > - **W2 (Non-uniqueness of constitutions):** We added an in-depth discussion of this limitation in Appendix G, referenced in the limitations section.
> > - **W3 (Practical applications):** We included a further use-case study on scaling annotation data in Appendix C.3 to complement the additional experiments, demonstrating ICAI's utility in real-world settings. This is in addition to the previously added bias detection experiments.
> > - **W4 (Bias amplification risk):** We further expanded the ethics statement and added warnings to the code output to highlight the risks of bias amplification and provide actionable mitigation steps.
> >
> > We hope these changes, along with those already made, adequately address your main concerns. We certainly believe they strengthen the manuscript and welcome any additional feedback.

---

> > > ### Author Response · Authors · 2024-12-02
> > >
> > > Thank you again for taking the time to review our work and for engaging in this constructive discussion! As a gentle reminder, the discussion period is ending soon (with less than a day remaining for reviewer comments). Does our latest response address your remaining concerns, or are there any aspects we could clarify further?

---

### Official Review · Reviewer_dHpo · 2024-11-04

**Soundness:** 2
**Presentation:** 4
**Contribution:** 3
**Rating:** 8
**Confidence:** 4

**Summary:**

This paper proposes a novel and interesting problem, namely, inverse constitutional AI (ICAI) problem which aims to reconstruct the preference data based on some principles that are in reverse concluded from the preference data.

As an initial algorithm, the ICAI method involves prompting the LLM to generate the principles that summarize the preference patterns within the data. These principles are cleaned via clustering, deduplication, and testing by reconstruction loss, relevance, as well as credit ordering,

The experiments on diverse tasks and settings demonstrate its effectiveness.

**Strengths:**

- Very interesting and well-defined research problem.
- The ICAI method is simple and effective.
- The experiments cover various settings, including population preference, persona-based preference, and even personalized preference.

**Weaknesses:**

1. Static principles (with limited quantity) may lead to some information loss for summarizing the preference patterns. The number of patterns does matter. For example, in the paper of PopAlign[1], the authors have investigated the so-called elicitive contrast for preference data synthesis, which involves generating good v.s. bad principles for each instruction as the thoughts for contrastive response generation. Such dynamic (or instruction-dependent) principles may benefit from the unlimited expressivity. Thus, as one more baseline, can the author add the elicitive preference annotation method, which involves generating principles for each instruction in an online manner as the thoughts for feedback labeling (instead of generating limited principles in an offline manner)?
2. The comparison between default feedback annotators and constitution-based feedback annotators on the unaligned settings may be unfair. Since default annotators are prompted to label the normal feedbacks, while the constitution-based annotators are prompted to label the special feedbacks. Do you prompt the default annotators to flip the feedbacks?
3. Once again, principles (in natural language) may lead to some information loss for summarizing the preference patterns. In contrast, a reward model can capture the preference patterns in an implicit “language” (i.e., model parameter) form. Can the authors add a reward model such as a fine-tuned PairRM[2] as one additional baseline?

[1] **PopAlign: Diversifying Contrasting Patterns for a More Comprehensive Alignment** https://arxiv.org/abs/2410.13785

[2] [llm-blender/PairRM · Hugging Face](https://huggingface.co/llm-blender/PairRM)

**Questions:**

1. Rule-based reward models are proven to be quite useful for the safety aspect [3]. Can the author compare the effects of the ICAI methods on different aspects? For example, helpful v.s. harmless?
2. Typos:
    - line 1088: the word “is” is redundant.

[3] Rule Based Rewards for Language Model Safety, OpenAI, [cdn.openai.com/rule-based-rewards-for-language-model-safety.pdf](https://cdn.openai.com/rule-based-rewards-for-language-model-safety.pdf)

---

> ### Author Response · Authors · 2024-11-20
>
> Thank you for your detailed review! Below, we discuss each point raised. We abbreviate weakness with "W" and question with "Q". All references refer to the revised manuscript, in which we highlight changes in blue.
>
> ### W1: Static principles may lead to information loss compared to dynamic methods like PopAlign. Can the authors include an elicitive preference annotation as a baseline?
>
> Thank you for highlighting this work. We politely note that the "PopAlign" paper was published over two weeks (17 Oct 2024) after the ICLR submission deadline. Not including a baseline that was unavailable at the time of submission should, in our opinion, neither be considered a weakness of our work nor affect the review score.
>
> That said, we appreciate the relevance of PopAlign and its elicitive contrast approach, which dynamically derives instruction-specific principles for generating diverse responses. While this approach shares similarities with our principle generation method, our understanding is that PopAlign is designed for data *synthesis* (response generation) rather than the *data interpretation* setting considered in our work. Applying this method directly to interpret existing datasets would require significant adaptation, as PopAlign does not incorporate responses and preferences into its principle generation process, nor does it evaluate principles for global applicability across multiple data points. A straightforward adaption would be similar to a chain-of-thought approach, possibly enhancing the default annotator performance, but unable to dynamically adjust to different preference datasets in a data-driven manner.
>
> To acknowledge this complementary work, we have added a discussion to Appendix B.3. We note how ICAI's data-driven constitutions could inspire more targeted contrastive prompts for PopAlign, and how PopAlign's strategies for generating diverse responses could enrich preference data for ICAI, advancing the shared goal of understanding and leveraging preferences to improve AI alignment. We appreciate that you brought this connection to our attention.
> Please let us know in case we misunderstood the PopAlign method or its applicability to the ICAI problem.
>
> More generally, we do agree that the Inverse Constitutional AI approach represents a compression of preference patterns. Yet, this compression is by design and allows us to generate short, interpretable principles and constitutions. Less compressed methods, such as conventional reward models, lack the interpretability features of our method. We have added a discussion of this trade-off to Section 6 (limitations) to clarify this point.
>
> ### W2: The comparison between default and ICAI annotators may be unfair. Did the authors prompt default annotators to flip feedbacks?
>
> We disagree that the comparison between our default and ICAI annotators is unfair but recognize that further clarification would be helpful. The critical difference between our annotator and the conventional LLM-as-a-judge annotators (such as the default baseline) is that our method can *adapt* to new datasets. Based on a (potentially small) training annotation set, our method dynamically generates an interpretable constitution and generates similar annotations. We could also compare our method to a conventional LLM-as-a-judge model that is prompted to select the worse output. However, such a "flipped" LLM-as-a-judge would then fail in the aligned setting. Our ICAI method can adapt in *both* scenarios to successfully reconstruct annotations. We have added a clarification to the beginning of Section 4 to address this point.
>
>
> ### W3: Principles in natural language lose information compared to implicit preference encoding in reward models. Can a reward model such as a fine-tuned PairRM be added as a baseline?
>
> We appreciate the suggestion to compare ICAI to a reward model such as PairRM and agree that reward models can capture preference patterns in an implicit form, which can be advantageous for certain applications. However, reward models lack the interpretability of natural language principles, which can be crucial for understanding and debugging model behaviour. ICAI's strength lies in its ability to provide human-readable explanations of preference patterns, which can be valuable for model transparency and user trust. We have extended the limitations section (Section 6) to discuss this trade-off, highlighting the interpretability of ICAI's principles compared to the implicit nature of reward models. We hope this clarifies the motivation behind our choice of method and baselines.

---

> ### Author Response · Authors · 2024-11-20
>
> ### Q1: Rule-based reward models are proven to be quite useful for the safety aspect. Can the author compare the effects of the ICAI methods on different aspects? For example, helpful v.s. harmless?
>
> Again, we would like to note that the "Rule Based Rewards for Language Model Safety" paper was first [submitted to arXiv](https://arxiv.org/abs/2411.01111v1) on November 2nd, roughly one month after the ICLR submission deadline, and could therefore not be considered in our initial submission. Nonetheless, we appreciate that you brought this work to our attention, as it indeed seems quite related. The authors of the RBR paper propose a method to learn an auxiliary safety reward model that composes natural-language "propositions" (generally binary statements about the response, e.g., "contains an apology") using a linear combination of these propositions as a reward signal. These propositions have some resemblance to our principles, but are hand-crafted and only used as features, not as direct preference indicators. Exploring a similar approach of propositions combined with a thin reward model could be an interesting alternative approach for the ICAI problem, also allowing for comparatively cheap fine-tuning with a manually modified constitution.
>
> The authors of the RBR paper further express a preference for detailed rules over vague principles such as "prefer the helpful response" for steerability and interpretability reasons, which could be complemented by an ICAI-like approach to generate candidate rules in a data-driven way. We have added a discussion of this connection to Appendix B.3 of the paper, highlighting the potential interaction between the two methods. We hope this clarifies the relationship between our work and the Rule Based Rewards paper.
>
> Additionally, we started to explore our interaction with helpful and harmful aspects on Anthropic's [HH-RLHF dataset](https://github.com/anthropics/hh-rlhf). We are still working on those experiments and plan to share them once they are completed.
>
> ### Q2: Typos: line 1088: the word “is” is redundant.
>
> Thank you for finding this typo! We have now fixed this prompt in our codebase but would not expect this redundant "is" to have affected our results in any notable way.

---

> ### Comment · Reviewer_dHpo · 2024-11-27
> **Reviewer Response**
>
> Thanks for your clarifications. But there are some misunderstandings.
>
> Q1: In ICAI, the workflow is: (1) generate principles from the "train" data, (2) generate preference labels for a "test" sample according to the **fixed** principles. While the workflow of an elicitive approach is: (1) generate principles for a "test" sample, (2) generate preference labels for this "test" sample according to these **generated** principles (no "train" data).
>
> Q2: A flipped LLM-as-a-judge means: If the LLM prefers one response, the preference label is "rejected"; if it rejects a response, the preference label is "preferred".
>
> Q3: I am still curious about the reward model performance despite its lack of interpretability.
>
> Q4: I am convinced. Thank you.
>
> This paper presents interesting ideas, but it would be significantly improved by including more baselines. I am willing to increase my score to 6 if at least one of these baselines is thoroughly discussed. Furthermore, if all these baselines are incorporated and analyzed, I would consider increasing my score to 8.

---

> > ### Author Response · Authors · 2024-11-28
> >
> > Thank you for your feedback and for clarifying your points. In response, we have carefully revised our paper to incorporate the suggested baselines, along with extensive discussion, as detailed in Section 4 and Appendix F. New changes over the previous revision are highlighted in **green**. We added the following baselines:
> >
> > - **Default (flipped)**: A variation of the original Default baseline where preference labels are systematically inverted (following the method suggested). This enables a more rigorous evaluation of the annotator's performance under different alignment scenarios.
> >
> > - **PopAlign**: An adaptation of the PopAlign method for our pairwise preference annotation framework. This baseline generates instruction-specific principles for each response pair, offering a dynamic alternative to our static principle generation approach.
> >
> > - **PairRM**: Integration of the Pairwise Reward Model (PairRM) by Jiang et al. as a black-box preference model. PairRM jointly encodes response pairs and instructions to produce comparative quality scores.
> >
> > - **PairRM (tuned)**: A version of PairRM fine-tuned on our training data. This baseline allows us to explore the performance gains from domain-specific adaptation of a reward model. We found PairRM uniquely suited for this comparison, as it provides preference predictions on-par with the Default annotator while allowing for resource-efficient fine-tuning. We appreciate your suggestion of this model.
> >
> > We believe these additional baselines enhance the evaluation of our Inverse Constitutional AI (ICAI) method by providing additional context and highlighting the unique strengths of ICAI, particularly its interpretability and sample-efficient adaptability.
> >
> > **Other improvements:** Following your previous feedback, we have also applied our method to preference data focusing on safety aspect, in particular Anthropic's [HH-RLHF dataset](https://github.com/anthropics/hh-rlhf). These results demonstrate the use of our approach for annotation scaling in this area and are discussed in Appendix C.3.
> >
> > We hope these modifications address your concerns and provide deeper insights into our method. We welcome any further suggestions and look forward to your assessment of the revised paper.

---

> > > ### Author Response · Authors · 2024-12-02
> > >
> > > Thank you again for taking the time to review our work and for engaging in this constructive discussion! As a gentle reminder, the discussion period is ending soon (with less than a day remaining for reviewer comments). Do you have any final concerns or questions?

---

> > > > ### Comment · Reviewer_dHpo · 2024-12-03
> > > >
> > > > Thank you for your valuable updates. I am willing to improve my score to 8; however, I still have a few concerns:
> > > >
> > > > The presentation requires further revision to more clearly highlight the additional baselines.
> > > >
> > > > Regarding the PopAlign baseline, it would be more accurate to refer to it as "elicitive CAI" rather than PopAlign, since the new baseline is inspired by PopAlign's Elicitive Contrast, rather than the PopAlign method itself.

---

> > > > > ### Author Response · Authors · 2024-12-03
> > > > >
> > > > > Thank you for taking the time to consider our response and update your review!
> > > > >
> > > > > We appreciate your suggestion to clarify the naming of the PopAlign inspired baseline. We will ensure that this distinction is made clear in the final version of the paper. We will also revise the presentation to more clearly highlight the additional baselines. We are grateful for your feedback and your engagement in the discussion, and look forward to addressing these points in the camera-ready version of the paper.

---

### Official Review · Reviewer_VDd6 · 2024-11-04

**Soundness:** 2
**Presentation:** 2
**Contribution:** 3
**Rating:** 6
**Confidence:** 4

**Summary:**

This paper introduces the Inverse Constitutional AI (ICAI) problem, which seeks to generate a set of principles from a given feedback dataset. These principles serve as a concise and human-readable representation of the feedback dataset, potentially aiding in identifying annotation biases and scaling up feedback annotation. The authors propose an initial ICAI algorithm and evaluate it on four different feedback datasets. Results indicate that the summarized principles can assist large language models in reconstructing the feedback annotations.

**Strengths:**

1. The paper introduces a new problem, named Inverse Constitutional AI (ICAI), which aims to compress human or model feedback into principles that can help uncover biases in data annotation, enhance understanding of model performance, scale feedback to unseen data, and adapt large language models to individual or group preferences.

2. The paper proposes a straightforward method to address ICAI problems and conducts extensive experiments across four different feedback datasets to validate its approach.

3. The authors present a method to evaluate the effectiveness of the generated principles by inputting them into an LLM and requiring the model to reconstruct the original feedback datasets, with the agreement serving as an evaluation metric for the summarized principles.

**Weaknesses:**

1. The experimental results would be more convincing if the authors demonstrated the application of ICAI. For instance, providing experimental evidence of ICAI’s potential in addressing annotation biases and scaling up annotation would strengthen the paper. While the authors claim their algorithm can help discover annotation bias in the feedback dataset, the experiments focus solely on reconstructing the original feedback without analyzing bias discovery and annotation scaling.

2. The proposed method has inherent limitations: (1) In the first step, the LLM generates principles based on single feedback, but some annotation biases and principles require synthesis from multiple feedbacks. (2) In the second step, K-means clustering is used to group the generated principles, which requires specifying the number of clusters in advance. In real-world scenarios, the exact number of principles is usually unknown.

3. The experimental results could benefit from deeper analysis: (1) In Section 4.2, it is unclear why GPT-3.5-Turbo’s performance does not surpass random choice. Is this due to the quality of the generated principles, or does it reflect limitations in the model’s ability to reconstruct feedback from constitutions effectively? (2) In Sections 4.1 and 4.2, the default annotator cannot achieve better agreement than random choice. This requires further explanation. Does this suggest a bias in the preference data itself, or might the model be inherently biased?

**Questions:**

1. What are the distinctions between “best”, “medium”, and “worst” constitutions mentioned in Appendix H?

---

> ### Author Response · Authors · 2024-11-20
>
> Thank you for your thoughtful feedback! Below is a discussion where we aim to address each of your points. We abbreviate weakness with "W" and question with "Q". All references refer to the revised manuscript, in which we highlight changes in blue.
>
> ### W1: Limited discussion of applications of ICAI.
>
> Thank you for raising the points about expanding the discussion of ICAI's applications in addressing annotation biases and scaling up annotation. While our analysis of demographic differences in the PRISM dataset already demonstrates one practical application of ICAI, we acknowledge that the paper's primary focus has been on demonstrating our method's general capability via reconstruction performance, rather than broader applications. We view the reconstruction of feedback annotations as an essential first step, laying the foundation for future work on the broader applications of the ICAI method. We agree, though, that a deeper exploration of how ICAI can be applied would enhance the paper and are grateful for the feedback.
>
> **We have added further details on the application of ICAI for annotation scaling and bias detection based on existing and new experimental results.** We believe that ICAI's potential in scaling up annotation is already demonstrated by the generalization of the principles to *unseen test data* in the AlpacaEval experiments (Section 4.2 and Appendix C.6). The paper was previously lacking a discussion of this aspect, however, which we have now added to Section 4.2. We further address bias discovery by adding a case study in the new Appendix C.2, illustrating how ICAI can uncover annotation biases such as verbosity, lists, and assertiveness, and outlining potential extensions for detecting more subtle biases. For example, we observe that verbosity bias seems particularly prominent in the Chatbot Arena and PRISM datasets, while AlpacaEval and Chatbot Arena have a preference for assertive language ("a definitive stance without nuance"). These findings highlight ICAI's utility in uncovering and analysing dataset-specific biases.
> We hope these additions provide more comprehensive evidence for ICAI's applications and address your concerns effectively.
>
> ### W2 (1): Some annotation biases and principles require synthesis from multiple feedbacks.
>
> We agree that our current approach, generating principles based on a single preference pair, may miss principles that only become obvious when considering multiple preference pairs simultaneously. We chose the single preference to avoid additional complexity in our initial algorithm. To test whether this alternative approach has a notable effect on the performance of our method, **we add results of a new ablation study** with principle proposal via multi-preference prompts, see Appendix C.3. We observe mixed results: for some tasks, multi-preference proposal improves performance, for others it slightly decreases performance. A possible explanation may be that some datasets (e.g. synthetic orthogonal) are best reconstructed using specific rules that may be missed if looking for overarching principles on multiple preferences. Other tasks are best reconstructed with more general principles, more likely to be found when looking at multiple preferences. Overall, we consider further adaptations to the principle generation approach an interesting area for future study.
>
> ### W2 (2): The K-means clustering requires users to set a number of principles, even though the exact number of principles is unknown.
>
> We agree that the selected number of clusters can seem arbitrary. In practice, we found ourselves constrained in the number of clusters (and corresponding principles) we could practically test due to compute costs, rather than running out of different principles to test. For all but perhaps the small synthetic datasets, we found the model in Step 1 came up with more unique principles than we could test. Thus, in practice, the number of clusters is primarily determined by the compute budget. In general, if the compute budget is available, more clusters likely would not have a negative impact on performance as any less useful principles get filtered out after the testing and filtering steps. More clusters, and correspondingly more tested principles, increase the chance that a well-performing principle is found. Note we may want to limit the number of well-performing principles that are included in the final constitution, as we observed that longer constitutions lead to diminishing returns at some point (see Appendix C.4).

---

> > ### Author Response · Authors · 2024-11-20
> >
> > ### W3 (1): In Section 4.2, is GPT-3.5-Turbo's low performance due to limited principle quality or model capability?
> >
> > Excellent question! In the AlpacaEval experiments discussed in Section 4.2, all constitutions are generated by GPT-4o (see L287). Thus, the GPT-3.5-Turbo annotator's performance is likely due to the model's capabilities, not principle quality. An alternative hypothesis would be that the principles do not transfer between models. We tested this hypothesis in Figure 8 in Appendix C.5 (Constitution Transferability), where we run annotators based on Claude models with constitutions generated by GPT-4o. Unlike GPT-3.5-Turbo, the Claude models *are* able to outperform the random baseline using these constitutions. Overall, these results indicate that GPT-3.5-Turbo has limited capabilities to follow principles compared to these other models --- rather than inherent problems with the principles themselves. We have added a clarification to the discussion of the results in Section 4.2.
> >
> > ### W3 (2): Why does the default annotator underperform random choice in some experiments?
> >
> > The default annotator judges responses based on its inductive bias, which arises from its training data. In the unaligned and orthogonal experiments, the default annotator is trained on aligned data, where the annotator is expected to prefer the aligned response, leading to a systematic bias away from the "correct", unaligned response. In the synthetic orthogonal data, the principles were chosen not to strongly correlate with the LLM's inductive biases, leading to a roughly random performance. The principles were designed manually, however, and are not perfectly orthogonal, which explains the slight deviation from random performance. In the aligned cases, i.e., on datasets that align with the default annotator's inductive biases, the default annotator performs well above random, as expected. We have added a clarifying statement to the beginning of Section 4.
> >
> > ### Q1: What are the distinctions between “best”, “medium”, and “worst” constitutions mentioned in Appendix H?
> >
> > Unless otherwise stated, we generally use six different seeds in each of our experiments, resulting in six different constitutions.  To avoid bloating the document excessively, we only show a subset of these constitutions in the appendix.  To ensure that we provide a representative sample of the constitutions, we rank them based on their performance in the reconstruction task (accuracy). We then select the constitutions with the best, median, and worst performance for display in the appendix. We discuss this setup in the introductory paragraph of Appendix E and have adapted this paragraph for clarity following your question. Let us know if the process remains unclear or if we can provide further clarification.

---

> ### Comment · Reviewer_VDd6 · 2024-11-26
>
> Thanks for your detailed responses. I think most of my concerns have been addressed and I will raise my score to 6.

---

> > ### Author Response · Authors · 2024-12-02
> >
> > Thank you for taking the time to consider our response and updating your review! We appreciate your help in improving the paper.

---

### Author Response · Authors · 2024-12-03
**Rebuttal summary**

We again thank all reviewers for their helpful reviews and comments! We were excited to see the reviewers' genuine interest in our work, describing our problem as *important* (nRv6), *new* (VDd6), *very interesting* and *well-defined* (dHpo); our method as *powerful* and *novel* (HeqU), as well as *simple* and *effective* (dHpo); and our experiments as *diverse* (dHpo) and *extensive* (VDd6).

To conclude the discussion phase, we briefly summarize the main concerns raised in the reviews and the corresponding improvements we made to the paper.

---

**1. Application experiments** (VDd6, nRv6, dHpo)

VDd6 and nRv6 recommended including additional application experiments to better demonstrate the utility of our Inverse Constitutional AI (ICAI) method. In response, we have added extensive new experimental results focusing on two application use-cases: bias detection and annotation scaling. We conduct the scaling experiments on harmless/helpful data, as suggested by dHpo, to further explore our method's applicability to this domain.
*Changes: Section 4.5, Appendices C.2 and C.3*

---

**2. Additional baseline comparisons** (dHpo)

dHpo suggested adding results of additional baselines for better comparability. We have addressed this by including results for all suggested baselines (default flipped, PopAlign, reward model original/fine-tuned), strengthening our method's evaluation and showcasing its advantages in our problem domain relative to these baselines.
*Changes: Section 4, Appendices D.5 and F*

---

**3. Alternative principle testing** (VDd6)

VDd6 recommended evaluating a setup where principles are generated based on multiple preferences rather than individual ones, as in our method. We conducted an additional experiment to test this alternative approach and have included the results, which indicate mixed outcomes.
*Changes: Appendix C.4*

---

**4. Risks of misinterpretation and misuse** (nRv6)

nRv6 raised concerns regarding the risk associated with misusing or misinterpreting our method's results. Relatedly, dHPO and HeqU highlighted the possibility of a short constitution missing nuances in annotator decisions. In response, we extended our paper's existing discussions on these topics as well as adding warnings to our code, to ensure that users are aware of and can mitigate these limitations.
*Changes: Section 6, Ethics Statement, Appendix G*

---

**5. Additional revisions**

In addition to the major concerns outlined above, we have addressed several smaller comments and suggestions from the reviewers. To keep this summary concise, we refer to the original reviews and responses for a comprehensive list.

---

Overall, we believe that the reviewers' suggestions have helped us to substantially improve our paper. We again thank all reviewers for their efforts!

---

### Meta-Review · Area_Chair_PKj3 · 2024-12-18

**Metareview:**

**Summary:**

This paper highlights the limitations of the current pairwise feedback data in preference optimization. We only know which one is better than another, but don't know "why". So, the author introduces a new problem called an ICAI problem, which formulates the interpretation of pairwise text preference data as a compression task. And the validation process of this task is done by checking whether we can reconstruct the original human feedback based on the constitutions. The algorithm follows five steps from principle generation to principle filtering, where most of the parts seem to rely on the use of LLMs. The experiments demonstrate that the effectiveness of the proposed algorithm for four types of datasets, including synthetic data, AlpacaEval, Chatbot Arena data, and PRISM.

While reviewing the feedback from the reviewers, I have identified several key strengths and weaknesses of the paper.

**Strength:**
- Addresses a critical challenge in AI research by developing interpretable constitutional principles for preference learning.
- Introduces the novel problem of Inverse Constitutional AI (ICAI) with clear applications, including uncovering biases, improving model performance understanding, and adapting models to diverse preferences.
- Converts opaque preference data into clear, natural language principles, enhancing understanding of model behavior and building trust in AI systems.
- Offers a powerful tool for identifying systematic biases in human-annotated feedback, enabling more balanced training processes and better-aligned models.
- Includes experiments on population preferences, persona-based preferences, and personalized preferences, showcasing broad applicability.

**Weakness:**
- While the paper claims ICAI can address annotation biases and scale up annotation, no experiments demonstrate these capabilities.
- Unclear why GPT-3.5-Turbo's performance is no better than random choice in Section 4.2—further analysis is needed.
- The comparison between default feedback annotators and constitution-based annotators may be unfair due to differing prompt settings.
- Summarizing preference patterns into static principles may oversimplify complex data, resulting in loss of nuance that dynamic methods or reward models could better capture.

**Additional Comments On Reviewer Discussion:**

The authors participated in rebuttal and the provided clarification mostly cleared the initial concerns of the reviewers.

**Decision:**

I believe the strengths of this paper far outweigh the weaknesses highlighted by the reviewers. Two reviewers awarded very high scores of 8, and even the borderline score of 5 given by reviewer nRv6 was accompanied by the remark, "I believe they adequately reflect the quality and contribution of your work." To summarize, I recommend "accept (spotlight)" as this paper addresses a novel and significant task with the potential for high impact in the field of preference optimization.

---

### Decision · Program_Chairs · 2025-01-22

Accept (Poster)